**1  Triple oxygen isotope evidence for the pathway of nitrous oxide**

**2  production in a forested soil with increased emission on rainy days**

Weitian Ding[1], Urumu Tsunogai[1], Tianzheng Huang[1], Takashi Sambuichi[1], Wenhua
Ruan[1], Masanori Ito [1], Hao Xu[1], Yongwon Kim [2], Fumiko Nakagawa[1]
[1]Graduate School of Environmental Studies, Nagoya University, Furo-cho, Chikusa-ku,
Nagoya 464-8601, Japan
[2]International Arctic Research Center, University of Alaska Fairbanks, Fairbanks, Alaska
99775-7320, USA
Corresponding author: Weitian Ding
Email: dwt530754556@gmail.com

**Abstract**

Continuous increases in atmospheric nitrous oxide ($N_2O$) concentrations are a global concern. Both nitrification and denitrification are the major pathways of $N_2O$ production in soil, one of the most important sources of tropospheric $N_2O$. The $^{17}O$ excess ($\Delta^{17}O$) of $N_2O$ can be a promising signature for identifying the main pathway of $N_2O$ production in soil. However, reports on $\Delta^{17}O$ are limited. Thus, we determined temporal variations in the $\Delta^{17}O$ of $N_2O$ emitted from forested soil for more than one year and that of soil nitrite ($NO_2^-$), which is a possible source of O atoms in $N_2O$. We found that $N_2O$ emitted from the soil exhibited significantly higher $\Delta^{17}O$ values on rainy days ($+0.12\pm0.13$ ‰) than on fine days ($-0.30\pm0.09$ ‰), and the emission flux of $N_2O$ was significantly higher on rainy days ($38.8\pm28.0$ µg N $m^{-2}h^{-1}$) than on fine days ($3.8\pm3.1$ µg N $m^{-2}h^{-1}$). Because the $\Delta^{17}O$ values of $N_2O$ emitted on rainy and fine days were close to those of soil $NO_2^-$ ($+0.23\pm0.12$ ‰) and $O_2$ ($-0.44$ ‰), we concluded that although nitrification was the main pathway of $N_2O$ production in the soil on fine days, denitrification became active on rainy days, resulting in a significant increase in the emission flux of $N_2O$. This study reveals that the main pathway of $N_2O$ production can be identified by precisely determining the $\Delta^{17}O$ values of $N_2O$ emission from soil and by comparing the $\Delta^{17}O$ values with those of $NO_2^-$, $O_2$, and $H_2O$ in the soil.

## 1. Introduction

Nitrous oxide ($N_2O$) is a strong greenhouse gas and an essential substance in stratospheric ozone depletion (Dickinson and Cicerone, 1986). Since pre-industrial times, the atmospheric $N_2O$ level has increased by 24 % to 335.8 ppb, with an average growth rate of 1.05 ppb $yr^{-1}$ in the last decade (WMO, 2023). Terrestrial soils account for ~60 % of total $N_2O$ emissions (Tian et al., 2020). Therefore, better knowledge of the pathways of $N_2O$ production in soils is required to establish mitigation measures.

Both nitrification and denitrification are representative microbial pathways of $N_2O$ production in soils (Wrage et al., 2001). Nitrification is the oxidation of ammonium ($NH_4^+$) to nitrate ($NO_3^-$) via aerobic microbial activity, during which $N_2O$ is produced as a byproduct of hydroxylamine ($NH_2OH$) oxidation to nitrite ($NO_2^-$), while denitrification is the reduction of $NO_3^-$ to $NO_2^-$ and then to $N_2O$ which is further reduced to nitrogen ($N_2$) via facultative anaerobes (Figure 1). Soil conditions such as moisture content, $O_2$ availability (Bateman and Baggs, 2005; Zhu et al., 2013), temperature (Luo et al., 2007), and fertilizer types (Zhu et al., 2013) have been proposed as parameters to determine the pathways of $N_2O$ production in soils.

Techniques such as acetylene blockage (Balderston et al., 1976; Lin et al., 2019), artificial isotope tracers ($^{15}N$ and $^{18}O$) (Mulvaney and Kurtz, 1982; Wrage et al., 2004), and natural stable isotopes (Toyoda et al., 2013; Yu et al., 2020) are conventionally used to identify the pathways of $N_2O$ production via nitrification and denitrification. Both acetylene blockage and artificial isotope tracers are mostly performed in laboratory (*in vitro*) incubations because they are costly, complicated, and time-consuming in field research. Natural stable isotopes such as $\delta^{15}N$, $\delta^{18}O$, and SP ($^{15}N$ site preference) can be

used to identify the pathways of $N_2O$ production in soils (Decock and Six, 2013; Toyoda
et al., 2017; Verhoeven et al., 2019). However, further reduction of $N_2O$ to $N_2$ after the
production of $N_2O$ until emission from soil to air results in significant changes in the
$\delta^{15}N$, $\delta^{18}O$, and SP values of $N_2O$ due to the fractionation of isotopes, which makes the
identification process difficult (Ostrom et al., 2007).
Recent studies on the $\Delta^{17}O$ value of $NO_3^-$ (the definition detailed in Section 2.4) have
reported that $\Delta^{17}O$ is a useful natural signature for clarifying the complicated
biogeochemical processes in terrestrial ecosystems (Ding et al., 2022, 2023, 2024;
Michalski et al., 2004; Tsunogai et al., 2010). Although the values of $\delta^{15}N$, $\delta^{18}O$, and SP
can vary during various fractionation processes of isotopes within terrestrial ecosystems,
the $\Delta^{17}O$ value remains almost stable because possible variations in $\delta^{17}O$ and $\delta^{18}O$ values
during the processes of biogeochemical isotope fractionation follow the relation of $\delta^{17}O \approx$
$0.5\ \delta^{18}O$, which cancels out the variations in the $\Delta^{17}O$ value (Young et al., 2002).
Consequently, the mixing of the same oxygen compounds with different $\Delta^{17}O$ values is
the primary cause of variations in $\Delta^{17}O$ values throughout the biogeochemical processes
in terrestrial ecosystems.
Because $N_2O$ produced through nitrification is a byproduct of the oxidation reaction
between $NH_4^+$ (to $NH_2OH$) and $O_2$, the $\Delta^{17}O$ value of $N_2O$ produced through nitrification
is expected to be close to that of tropospheric $O_2$ (Figure 1) (Kool et al., 2007, 2011;
Wrage et al., 2005), with previous studies reporting a $\Delta^{17}O$ value of $-0.44$ ‰ (Sharp and
Wostbrock, 2021). Conversely, the $\Delta^{17}O$ value of $N_2O$ produced through denitrification is
expected to be close to that of $NO_2^-$ (Figure 1) (Kool et al., 2007, 2011; Wankel et al.,
2017; Wrage et al., 2005). Because O atoms in $NO_2^-$ are derived from either soil $NO_3^-$
($\Delta^{17}O$ = from 0 to +20 ‰) or $H_2O$ ($\Delta^{17}O$ = +0.03±0.01 ‰) (Hattori et al., 2019;
Nakagawa et al., 2018; Uechi and Uemura, 2019), significant differences in $\Delta^{17}O$ values
between $N_2O$ produced through nitrification and that produced through denitrification are
expected if the additional contributions of O atoms derived from soil $H_2O$ are
insignificant in $N_2O$ during the processes of $N_2O$ production in soils through nitrification
and denitrification (Figure 1) (Kool et al., 2007).

Previous studies have identified the elevated $\Delta^{17}O$ values in atmospheric $N_2O$ ($\Delta^{17}O \approx$

+0.9 ‰), observed in both stratospheric and tropospheric air (Cliff et al., 1999; Kaiser et
al., 2003; Thiemens and Trogler, 1991). Komatsu et al. (2008) subsequently conducted
the first $\Delta^{17}O$ measurements of $N_2O$ emitted from a soil to assess whether soil $N_2O$ could
be the source of elevated $\Delta^{17}O$ values of atmospheric $N_2O$. However, the temporal
variations of the $\Delta^{17}O$ values for $N_2O$ emitted from soil remain unknown. Besides,
whether $\Delta^{17}O$ values of $N_2O$ can be used to identify the pathways of $N_2O$ production in
soils has not been discussed. Additionally, the advantages of $\Delta^{17}O$ signature, relative to
other natural stable isotopes, for identifying the pathways of $N_2O$ production remain
unclear. To address these, in this study, we measured precise $\Delta^{17}O$ values for $N_2O$
emitted from forested soil and those for $NO_2^-$ in the soil.  Additionally, we conducted
similar observations in the same soil artificially fertilized with Chile saltpeter or urea to
investigate the possible contributions of O atoms derived from soil $H_2O$ in $N_2O$ during
$N_2O$ production.

**2. Methods**
**2.1 Study site**
The study site was located in a secondary warm-temperate forest within an urban area
(35°10'N, 136°58'E, Figure 2), approximately 50 m from the common building of the
Graduate School of Environmental Studies at Nagoya University. The lowest, highest,
and mean monthly temperatures recorded at the nearest meteorological station (Nagoya
station) were 5.2 °C (in January), 28.9 °C (in July), and 18.5 °C, respectively, from April
2022 to July 2023. The annual mean precipitation was approximately 1800 mm. The soil
stratum in the forested field possessed an approximate depth of 20 cm, characterized by a
bulk density of 1.12 $g/cm^3$. Details of the forest have been described in the previous study
(Hiyama et al., 2005).

**2.2 Sampling of $N_2O$**
Samples of $N_2O$ emitted from the forested soil under natural conditions were collected
18 times (n = 18) from April 2022 to July 2023 in a field with an area of 5 $m^2$ (Figure
2b). Among the samples, 12 were collected on fine days, whereas 6 were collected on
rainy days. A fine day is defined as a day without precipitation for 48 hours prior to the
end of each sampling. The total precipitation within 12 h at the end of each sampling of
the rainy days exceeded 12 mm.
The sampling of $N_2O$ emitted from the artificially fertilized soil was performed during
a period of fine weather in three plots (1 $m^2$ for each located more than 5 m away from
each other) within the same forested field, located approximately 3 m away from the plot
where we conducted the sampling under natural conditions (Figures 2b and 2c). Either
urea ($CO(NH_2)_2$, 46 % TN) or Chile saltpeter ($KNO_3$, 14 % TN) was applied to two of
the plots (U and CS plots) on 2023/7/16 at the same N amount of 250 kg N $ha^{-1}$. Urea is a
synthetic N fertilizer (Sun & Hope Ltd., Japan), and Chile saltpeter (SQM Ltd., USA)
contains $NO_3^-$ with a high $\Delta^{17}O$ value of +19 ‰ (determined through the internationally
distributed isotope reference materials USGS-34 and USGS-35). The third plot was
blank, meaning no fertilizer was added (NF plot). Sampling of $N_2O$ from each plot was
performed twice on days 2 and 6 after the addition of each fertilizer.
To precisely determine $\Delta^{17}O$ of $N_2O$, more than 60 nmol of $N_2O$ is required (Komatsu
et al., 2008), which corresponds to more than 4 L of air containing $N_2O$ at atmospheric
concentrations. Accordingly, in this study, a flow chamber made of polypropylene with
dimensions of 0.8 m × 0.3 m × 0.18 m was deployed onto the sampling site throughout
each day of sampling (Figure S1). This chamber has an inlet and outlet port with an inner
diameter of 1 cm. The outlet port was connected to an air pump using Tygon tubing, and
the inlet port was open to ambient air. Using the air pump, the air in the chamber was
taken into a 5-L aluminum bag, along with the gases emitted by the soil, as illustrated in
Figure S1. The flow rate of the air pump was set at 100 ml/min throughout the
deployment of the chamber; thus, each sampling lasted 45 min until 4.5 L of gas was
collected into the aluminum bag. Each gas sampling was started 2 h after deployment of
the flow chamber; thus, it took more than 8 h to collect four samples. In addition to the
gas samples emitted from the soil, ambient air in the forest was sampled into two 3-L
vacuum stainless steel canisters (SilcoCan, Restek).

**2.3 Sampling and analysis of forested soil**
After collecting the gas samples to determine $N_2O$, a soil sample (approximately 150
g) was randomly collected from more than four places beneath the chamber.
Approximately 20 g of the soil sample was heated at 80 °C for 48 h to estimate the water
content from the weight loss and water-filled pore space (WFPS; the calculation was
detailed in Text S1). Using the remaining soil sample (120 g), $NH_4^+$, $NO_3^-$, and $NO_2^-$ in
each soil sample were extracted into 120 mL of a 2-M KCl solution, and their
concentrations were determined using a high performance microflow analyzer (QuAAtro
39 Autoanalyzer, BLTEC, Osaka, Japan).

**2.4 Concentration and isotopic compositions of $N_2O$**
The gas samples collected in aluminum bags or stainless canisters were subsampled
into a 100-ml pre-evacuated glass bottle to determine the concentration ($[N_2O]$), $\delta^{15}N$,
and $\delta^{18}O$ of $N_2O$ simultaneously. The remaining samples were further subsampled to
either 1 or 2 L pre-evacuated glass bottles to determine the $\Delta^{17}O$ of $N_2O$. The
concentration and isotopic compositions ($\delta^{15}N$, $\delta^{18}O$, and $\Delta^{17}O$) of $N_2O$ were determined
using a continuous flow isotope ratio mass spectrometry (CF-IRMS; Finnigan MAT252,
Thermo Fisher Scientific, Waltham, MA, USA) system that consists of an original pre-
concentrator system, chemical traps, and gas chromatograph at Nagoya University
(Komatsu et al., 2008). The analytical procedures using the CF-IRMS system were the
same as those detailed in previous studies (Hirota et al., 2010; Komatsu et al., 2008).
The isotopic ratios of $^{15}N/^{14}N$, $^{17}O/^{16}O$, and $^{18}O/^{16}O$ are expressed in the $\delta$ notations:
$$\delta^{15}N, \delta^{17}O, \text{ or } \delta^{18}O = R_{sample}/R_{standard} - 1 \tag{1}$$
where R denotes $^{15}N/^{14}N$, $^{17}O/^{16}O$, or $^{18}O/^{16}O$ ratios of the sample and each standard
reference material.
The $\Delta^{17}O$ of $N_2O$, including $NO_2^-$, $NO_3^-$, $H_2O$, and $O_2$, is defined by Eq. 2 (Kaiser et
al., 2007; Miller, 2002):
$\Delta^{17}O = \frac{1+\delta^{17}O}{(1+\delta^{18}O)^\beta} - 1$                                  (2)
where $\beta$ denotes the slope of the reference line in the $\delta^{17}O-\delta^{18}O$ space. Previous
studies have proposed values ranging from 0.525 to 0.5305 for $\beta$ during the various
processes of isotope fractionation through experimental measurements and/or theoretical
calculations (Cao and Liu, 2011; Matsuhisa et al., 1978; Pack and Herwartz, 2014; Sharp
and Wostbrock, 2021). In this study, we adopted a value of 0.528 for $\beta$ to define $\Delta^{17}O$.
The details of the ranges of the possible $\Delta^{17}O$ variations due to the ranges of $\beta$ are
presented in Section 4.1.
To calibrate the $\delta^{15}N$ and $\delta^{18}O$ of $N_2O$ to the international scale, $N_2O$ in a tropospheric
air sample collected at Hateruma Island in 2010 (Japan) was used as the standard with a
$\delta^{15}N$ value of +6.5 ‰ and a $\delta^{18}O$ value of +44.3 ‰ (Toyoda et al., 2013). To calibrate the
$\Delta^{17}O$ of $N_2O$ on the international VSMOW (Vienna Standard Mean Ocean Water) scale,
we prepared two kinds of $N_2O$ standards with different $\Delta^{17}O$ values calibrated using a
conventional method (Thiemens and Trogler, 1991). The procedures for this calibration
are presented in Section 2.6, with the details of the $N_2O$ standards. Through repeated
measurements of $N_2O$ in a tropospheric air sample collected at Nagoya University, the
analytical precisions (1σ) of the measurements were estimated to be ±10.0 ppb, ±0.5 ‰,
±0.6 ‰, and ±0.11 ‰ for concentration, $\delta^{15}N$, $\delta^{18}O$, and $\Delta^{17}O$, respectively (Figure S2).
To achieve higher precision, analyses of $\Delta^{17}O$ were performed at least three times for
each sample, resulting in a standard error (SE) of ±0.06 ‰.

## 2.5 Emission flux


Based on the change in the concentration of $N_2O$ from the inlet to the outlet, the
emission flux of $N_2O$ from the soil was calculated using Eq. 3:
$$\text{Flux} = \frac{P \times V \times (C_{final} - C_{air}) \times M}{R \times T \times t \times A}$$    (3)
where Flux denotes the emission flux of $N_2O$ ($\mu g$ N $m^{-2}$ $h^{-1}$), P denotes the pressure (Pa),
V represents the volume of the gas sample in the aluminum bag (0.0045 $m^3$), $C_{final}$
denotes the concentration of $N_2O$ in the gas sample taken at the end of each deployment
of the chamber ($\mu mol$ $mol^{-1}$), $C_{air}$ denotes the concentration of $N_2O$ in the ambient air
($\mu mol$ $mol^{-1}$), M represents the molecular weight of N in $N_2O$ (28 $\mu g$ N $\mu mol^{-1}$), R
represents the universal gas constant (8.314 $m^3$ Pa $K^{-1}$ $mol^{-1}$), T represents the air
temperature in the forest (K), t represents the duration of each gas sampling (45 min), and
A represents the surface area of soil covered by the chamber (0.24 $m^2$).

## 2.6 Calibration of the $\Delta^{17}O$ values of $N_2O$


To determine the $\Delta^{17}O$ values of $N_2O$ in the samples on the VSMOW scale, we
prepared two standards (STD1 and STD2) containing $N_2O$. The $\Delta^{17}O$ values of $N_2O$ in
the standards were calibrated to the VSMOW scale using the conventional method
reported in (Thiemens and Trogler, 1991), where $N_2O$ was quantitatively converted to $O_2$
using $BrF_5$ and a Ni catalytic container. The details are presented below.
A calibrated quantity of $N_2O$ (50–170 $\mu mol$) was subsampled and transferred into a
nickel tube (approximately 60 $cm^3$) under liquid $N_2$ temperature. The coexisting
components of $N_2O$, such as helium in the case of STD2, were evacuated from the nickel
tube after $N_2O$ was trapped in the nickel tube under liquid $N_2$ temperature. The nickel

tube was then heated at 725 °C for 2.5 h to convert $N_2O$ to $NiO$ and $N_2$. After evacuating

$N_2$ from the nickel tube, a 10-fold quantity of $BrF_5$ was introduced into the nickel tube

and heated at 725 °C for 12 h to convert $NiO$ to $O_2$ and $NiF_2$. After the purification of $O_2$,

both $\delta^{18}O$ and $\Delta^{17}O$ of $O_2$ were determined on the VSMOW scale using IRMS, with the

quantity of $O_2$ evolved from $N_2O$. Details on the procedures of $O_2$ purification and the

measurement of $O_2$ using IRMS on the VSMOW scale have been described in previous

studies (Sambuichi et al., 2021, 2023). STD1 is pure $N_2O$ gas prepared from $N_2O$ in a gas

cylinder (more than 99.9 %; Koike Medical Ltd., Japan). The yield ratio of $O_2$ and $\Delta^{17}O$

of STD1 were $103\pm7$ % and $-0.22\pm0.07$ ‰, respectively (Figure S3). The $N_2O$ in STD2

is a mixture of helium and $N_2O$ ($N_2O/He \approx 1.5$) produced from $NO_2^-$ that had been under

oxygen isotope exchange equilibrium with $H_2O$ with a $\Delta^{17}O$ value of $+1.2$ ‰ originally,

under a pH of 1.2. $NO_2^-$ was then converted to $N_2O$ through a reaction with hydrazoic

acid ($N_3H$), as described by (Tsunogai et al., 2008). The reaction product ($N_2O$) was

purged from the vial using pure helium (more than 99.9 %). After the removal of $H_2O$ by

passing a trap under the temperature of dry ice + ethanol, $N_2O$ was captured in a trap at

the temperature of liquid $O_2$ and then transported into a 1-L stainless steel canister

together with helium. The yield of $O_2$ and $\Delta^{17}O$ of STD2 were $97\pm5$ % and

$+1.13\pm0.02$ ‰, respectively (Figure S3). To calibrate the $\Delta^{17}O$ values of the samples

measured using CF-IRMS, approximately 1 mL of each STD was subsampled into a 200-

mL pre-evacuated glass bottle and diluted using pure helium to 1 atm. The $\Delta^{17}O$ values of

$N_2O$ in the diluted standards were then determined using CF-IRMS like the procedure

used on the samples before the sample measurements by introducing 30–60 nmol of $N_2O$.

This allowed us to calibrate the $\Delta^{17}O$ values of the samples to the VSMOW scale (Figure
S4).

**2.7 Isotopic composition of $NO_2^-$**
To determine the $\delta^{18}O$ and $\Delta^{17}O$ values of soil $NO_2^-$ that had been extracted in the KCl
solution, the $NO_2^-$ in the KCl solution was chemically converted to $N_2O$ using the
method originally developed to determine the $\delta^{18}O$ of $NO_2^-$ (McIlvin and Altabet, 2005),
with several modifications for $\Delta^{17}O$ (Xu et al., 2021), as explained below. Approximately
40 mL of each solution was pipetted into a glass vial (66.7 mL) and sealed with a butyl
rubber septum cap. After purging the solution using high-purity helium for 45 min,
1.8 mL of an azide-acetic acid buffer (0.1 mol $L^{-1}$ $NaN_3$ in 1 vol. % acetic acid), which
had been purged using pure helium as well, was added to the solution to convert $NO_2^-$ to
$N_2O$:
$HNO_2 + HN_3 \rightarrow N_2O + H_2O + N_2$                    (R1)
After the vials were shaken for 1 h at a rate of 2 cycles $s^{-1}$, 0.9 mL of 6-M NaOH was
added to each vial and shaken for 15 min.
The $\delta^{18}O$ and $\Delta^{17}O$ of $N_2O$ converted from $NO_2^-$ in each vial were determined using
the CF-IRMS system. We repeated the analyses for each solution sample at least three
times to obtain better precision for $\Delta^{17}O$.
The $\delta^{18}O$ values of $NO_2^-$ were calibrated to the VSMOW scale using three in-house
nitrite standards (STD10, STD11, and STD12), the $\delta^{18}O$ values of which had been
determined using a thermal conversion/elemental analyzer IRMS system, where oxygen
atoms in each nitrite/nitrate had been converted into CO using a glassy carbon tube at
1400 °C (Xu et al., 2021) and calibrated to the VSMOW scale using the international
nitrate standards USGS34 ($\delta^{18}O = -27.9$ ‰) and IAEA-NO-3 ($\delta^{18}O = +25.6$ ‰) as the
primary standards. Isotope fractionations during chemical conversion into $N_2O$ were
corrected by measuring the nitrite standards in the same way as samples were measured
using the CF-IRMS system. In addition, the extent of oxygen isotope exchange between
$NO_2^-$ and $H_2O$ during the conversion was quantified using the relation between $\delta^{18}O$ of
the nitrite standards and that of $N_2O$ (Xu et al., 2021). The $\Delta^{17}O$ values of $NO_2^-$ were
calibrated to the VSMOW scale by comparing $N_2O$ derived from $NO_2^-$ with $N_2O$
standards (STD1 and STD2) while assuming that the changes in $\Delta^{17}O$ were negligible
during the conversion from $NO_2^-$ into $N_2O$, except for the oxygen isotope exchange
reaction between $NO_2^-$ and $H_2O$ during the conversion to $N_2O$. The progress of oxygen
isotope exchange between $NO_2^-$ and $H_2O$ was calibrated from the $\Delta^{17}O$ values of $NO_2^-$
using the exchange rate estimated by calculating $\delta^{18}O$ values while assuming that the
$\Delta^{17}O$ value of $H_2O$ was 0 ‰.

While the KCl solutions were widely used for the extraction of soil $NO_2^-$ (e.g.,

Lewicka-Szczebak et al., 2021; Shen et al., 2003), Homyak et al. (2015) raised the
concerns that the recovery of soil $NO_2^-$ could be low when using KCl solutions compared
to deionized water. Therefore, we conducted a comparative experiment to evaluate this
potential issue and concluded that the use of KCl solution introduced negligible bias in
terms of soil $NO_2^-$ recovery or $\Delta^{17}O$ measurements compared to deionized water
extraction for the soil type and experimental conditions in this study. The details are
described in the supplement (Text S2).

## 3. Results

### 3.1 Flux and isotopic compositions of $N_2O$ emitted from forested soil

Almost all of the concentrations of $N_2O$ ([$N_2O$]) in the samples collected in aluminum

bags were higher than that of $N_2O$ in ambient air (Figures 3a and S5), implying that $N_2O$

in the aluminum bags was a mixture of $N_2O$ in ambient air and $N_2O$ emitted from the

forested soil. To determine the isotopic compositions ($\delta^{15}N$, $\delta^{18}O$, and $\Delta^{17}O$) of $N_2O$

emitted from the soil, $N_2O$ derived from ambient air was excluded using the linear

correlation between 1/[$N_2O$] and the isotopic compositions ($\delta^{15}N$, $\delta^{18}O$, and $\Delta^{17}O$) during

mixing (Figures 3b, 3c, 3d, and S5), also was known as Keeling plot approach (Keeling,

1958; Tsunogai et al., 1998, 2003). This method assumes that the concentrations of $N_2O$

($N_2O/(N_2O + N_2)$) in the gases emitted from the soil were more than 3 %, allowing

1/[$N_2O$] to be approximated to be 0 (Text S3). The uncertainties associated with the

isotopic compositions of $N_2O$ emitted from soil (i.e., the intercept) were estimated by

applying the York method (Tsunogai et al., 2011; York et al., 2004) to the obtained

relationship between 1/[$N_2O$] as the independent variable and the isotopic compositions

as the dependent variable in which uncertainties of both independent and dependent

variables for individual data are considered.

The flux of $N_2O$ emitted from the forested soil determined on fine days varied from

$-0.2$ to 9.8 µg N m$^{-2}$ h$^{-1}$, with an average of 3.8±3.1 µg N m$^{-2}$ h$^{-1}$ (1SD; n = 12). In

addition, the emission flux during the warm seasons (from April to October; 5.1±2.8 µg

N m$^{-2}$ h$^{-1}$) was significantly higher than that during the cold seasons (from November to

March; 1.0±1.1 µg N m$^{-2}$ h$^{-1}$) (Figure 4a; Table S1), implying that the emission flux of

$N_2O$ on fine days exhibited clear seasonal variation. Furthermore, the average emission
flux of $N_2O$ determined on rainy days ($38.8\pm28.0$ µg N m$^{-2}$ h$^{-1}$; n = 6) was significantly
higher than that determined on fine days ($3.8\pm3.1$ µg N m$^{-2}$ h$^{-1}$) (Figures 4a and 4b).
These patterns of $N_2O$ emissions were in accordance with those of agricultural and
forested soils reported in previous studies (Anthony et al., 2023; Chen et al., 2012;
Choudhary et al., 2002; Yan et al., 2008).
Because of the small emission flux of $N_2O$ during the cold seasons, the linear
relationships between the isotopic compositions and $1/[N_2O]$ became insignificant in
some of the observations performed during the cold seasons (Figure S5, from Nov. 2022
to Jan. 2023). Thus, the uncertainties associated with the isotopic compositions estimated
for $N_2O$ emitted from the soil became enormous. Consequently, the isotopic
compositions of $N_2O$ emitted from the soil are not shown under the following conditions:
(1) the $[N_2O]$ in the gas sample collected at the end of each deployment of the chamber
did not exceed 130 % of that of ambient air, and (2) the linear correlation between
$1/[N_2O]$ and the isotopic compositions was statistically insignificant ($P > 0.05$). Similar
criteria have been adopted in previous studies (Kaushal et al., 2022; Opdyke et al., 2009).
The $N_2O$ emitted from the forested soil on fine days exhibited $\delta^{15}N$, $\delta^{18}O$, and $\Delta^{17}O$
values ranging from −27.5 ‰ to −17.9 ‰, from +26.1 ‰ to +37.6 ‰, and from −0.40 ‰
to −0.11 ‰, respectively, with average values and standard deviations (1SD) of
$-22.5\pm2.8$ ‰, $+30.9\pm4.3$ ‰, and $-0.30\pm0.09$ ‰, respectively (Figures 4g, 4e, and 4c).
On the other hand, $N_2O$ emitted from the forested soil on rainy days exhibited $\delta^{15}N$, $\delta^{18}O$,
and $\Delta^{17}O$ values ranging from −26.6 ‰ to −13.8 ‰, from +18.4 ‰ to +36.2 ‰, and from
−0.06 ‰ to +0.26 ‰, respectively, with average values and standard deviations (1SD) of
−20.4±5.0 ‰, +27.9±6.4 ‰, and +0.12±0.13 ‰, respectively (Figures 4g, 4e, and 4c).

The $NO_2^-$ exhibited $\delta^{18}O$ and $\Delta^{17}O$ values ranging from +2.4 ‰ to +12.0 ‰ and from

+0.04 to +0.50 ‰, respectively, with average values of +6.0±2.0 ‰ and +0.23±0.12 ‰,
respectively (n = 18, Figures 4e and 4c). These $\delta^{18}O$ values of $NO_2^-$ coincided well with
those determined in a previous study (Lewicka-Szczebak et al., 2021).

**3.2 Flux and isotopic compositions of $N_2O$ emitted from artificially fertilized soils**

The fluxes of $N_2O$ emitted from the NF (no fertilizer), U (fertilized with urea,

$CO(NH_2)_2$), and CS (fertilized with Chile saltpeter, $KNO_3$) plots were 5.2, 70.6, and
112.3 µg N $m^{-2}$ $h^{-1}$, respectively, 2 days after fertilization and 4.2, 56.7, and 39.4 µg N
$m^{-2}$ $h^{-1}$, respectively, 6 days after fertilization (Table S1). The fluxes of $N_2O$ emitted
from the U and CS plots were significantly higher than that from the NF plot, indicating
that the flux of $N_2O$ emitted from the soil increased significantly because of fertilization,
supporting the results reported in previous studies (Kaushal et al., 2022; McKenney et al.,
1978; Toyoda et al., 2011, 2017).

The $\delta^{15}N$, $\delta^{18}O$, and $\Delta^{17}O$ values of $N_2O$ emitted from the NF plot 2 days after

fertilization were −17.1±6.4 ‰, +36.1±6.7 ‰, and −0.37±0.20 ‰, respectively, whereas
those emitted from the NF plot 6 days after fertilization were −12.2±3.2 ‰,
+40.0±13.3 ‰, and −0.32±0.23 ‰, respectively. The $\delta^{15}N$, $\delta^{18}O$, and $\Delta^{17}O$ values of $N_2O$
emitted from the U plot 2 days after fertilization were −39.3±0.7 ‰, +34.4±0.4 ‰, and
−0.14±0.06 ‰, respectively, whereas those emitted from the U plot 6 days after
fertilization were −33.3±0.5 ‰, +25.7±0.6 ‰, and −0.16±0.05 ‰, respectively. The
$\delta^{15}N$, $\delta^{18}O$, and $\Delta^{17}O$ values of $N_2O$ emitted from the CS plot 2 days after fertilization
were $-19.3\pm0.6$ ‰, $+54.1\pm0.8$ ‰, and $+8.22\pm0.03$ ‰, respectively, whereas those
emitted from the CS plot 6 days after fertilization were $-11.3\pm0.7$ ‰, $+58.7\pm1.2$ ‰, and
$+7.36\pm0.17$ ‰, respectively (Figure 5). These flux, $\delta^{15}N$, and $\delta^{18}O$ of $N_2O$ emitted from
the NF, U, and CS plots correspond well with the results of many previous studies on
forested and artificial soils (or agricultural soils) (Kaushal et al., 2022; Kim and Craig,
1993; Snider et al., 2009; Toyoda et al., 2017; Wrage et al., 2004).

The $\delta^{18}O$ and $\Delta^{17}O$ values of $NO_2^-$ in the NF plot 2 days after fertilization were

$+2.7$ ‰ and $+0.42$ ‰, respectively, whereas those in the NF plot 6 days after fertilization
were $+1.3$ ‰ and $+0.35$ ‰, respectively. The $\delta^{18}O$ and $\Delta^{17}O$ values of $NO_2^-$ in the U plot
2 days after fertilization were $+7.6$ ‰ and $+0.31$ ‰, respectively, whereas those in the U
plot 6 days after fertilization were $+5.4$ ‰ and $+0.17$ ‰, respectively. The $\delta^{18}O$ and $\Delta^{17}O$
values of $NO_2^-$ in the CS plot 2 days after fertilization were $+29.0$ ‰ and $+8.26$ ‰,
respectively, whereas those in the CS plot 6 days after fertilization were $+45.2$ ‰ and
$+12.32$ ‰, respectively (Figure 6).

**4. Discussion**
**4.1 Identification of $N_2O$ production pathways in forested soil using $\Delta^{17}O$ signature**

Because O atoms in $N_2O$ emitted from soil can be derived from those in $NO_2^-$, $O_2$, or

$H_2O$ in soil (Figure 1), we can constrain the pathways of $N_2O$ production by comparing
the $\delta^{18}O$ and $\Delta^{17}O$ values of $N_2O$ with those of $NO_2^-$, $O_2$, and $H_2O$ in soil. Consequently,
we compiled the $\delta^{18}O$ and $\Delta^{17}O$ values of atmospheric $O_2$ ($+23.88$ ‰ for $\delta^{18}O$ and
$-0.44$ ‰ for $\Delta^{17}O$, (Sharp and Wostbrock, 2021)) and rainwater (ranging from $-2$ ‰ to
−10 ‰ for $\delta^{18}O$ in Japan, (Nakagawa et al., 2018; Takahashi, 1998; Uechi and Uemura,
2019; Zou et al., 2015); +0.03 ‰ for $\Delta^{17}O$ in Japan (Uechi and Uemura, 2019)), as
shown in Figures 4 and 6, along with those of soil $NO_2^-$ measured in this study.
The $\Delta^{17}O$ of $N_2O$ produced in the soil may differ from that of the source of O atoms
($O_2$, $NO_2^-$, $H_2O$) because of oxygen isotope fractionation during nitrification and
denitrification, as the value of $\beta$ in Eq. (2) may vary depending on the reactions. Thus,
prior to using $\Delta^{17}O$ values to identify the pathways of $N_2O$ production in soils, we
quantified the possible variations in the $\Delta^{17}O$ values of $N_2O$ during each reaction. The
details are presented below.
The fractionation of oxygen isotopes during the transformation of the O atoms in $O_2$ to
those in $N_2O$ through nitrification accompanies significant variations in the value of $\delta^{18}O$
from $O_2$ to $N_2O$ (Figures 4e and 6a). In addition to $\delta^{18}O$, the $\Delta^{17}O$ value of $N_2O$ produced
through nitrification could be somewhat different from that of $O_2$, even if all O atoms in
$N_2O$ were derived from $O_2$, due to the possible differences in $\beta$ from 0.528 during the
reaction (Figure 7). The average variation in $\delta^{18}O$ from $O_2$ to $N_2O$ due to nitrification
($\Delta\delta^{18}O$ ($N_2O-O_2$)) was estimated to be 9 ‰ on average (Figures 4e and 6a) based on the
difference in $\delta^{18}O$ values between $N_2O$ emitted from the soil in this study (+33±10 ‰; n
= 19) and $O_2$ in the literature (Sharp and Wostbrock, 2021). Conversely, we can expect
values from 0.525 to 0.5305 for $\beta$ in the various reactions (Cao and Liu, 2011; Matsuhisa
et al., 1978; Pack and Herwartz, 2014; Sharp and Wostbrock, 2021), where the $\beta$ of
nitrification may be included. Thus, we quantified the possible range of variations in the
$\Delta^{17}O$ value of $N_2O$ from that of $O_2$ to be less than 0.027 ‰ (Figure 7), based on the
observed $\Delta\delta^{18}O(N_2O-O_2)$ and the possible variation range of $\beta$.
Similarly, the fractionation of oxygen isotopes during the transformation of O atoms in
$NO_2^-$ to those in $N_2O$ through denitrification accompanies significant variations in $\delta^{18}O$
from $NO_2^-$ to $N_2O$ as well. The $\Delta^{17}O$ value of $N_2O$ produced through $NO_2^-$ reduction
could be somewhat different from that of $NO_2^-$, even if all O atoms in $N_2O$ were derived
from $NO_2^-$, due to the possible differences in $\beta$ from 0.528 during the reaction (Figure 7).
The average variation in $\delta^{18}O$ from $NO_2^-$ to $N_2O$ due to fractionation ($\Delta\delta^{18}O$
($N_2O-NO_2^-$)) was estimated to be 25 ‰ on average (Figures 4e and 6a) based on the
difference in $\delta^{18}O$ values between $N_2O$ (+33±10 ‰; n = 19) and $NO_2^-$ in this study
(+8±9 ‰; n = 24). Thus, we quantified the possible range of variations in the $\Delta^{17}O$ value
of $N_2O$ from that of $NO_2^-$ to be less than 0.075 ‰ (Figure 7), based on the observed
$\Delta\delta^{18}O$ ($N_2O-NO_2^-$) and the possible variation range of $\beta$, from 0.525 to 0.5305.
Similarly, kinetic fractionation during the reduction of $N_2O$ to $N_2$ accompanies
variation in $\delta^{18}O$ from original $N_2O$ to residual $N_2O$ as well. The $\Delta^{17}O$ value of residual
$N_2O$ could somewhat differ from that of the original $N_2O$. Previous studies have reported
the range of variations in $\delta^{18}O$ from original $N_2O$ to residual $N_2O$ due to kinetic
fractionation to be less than 10 ‰ on average through incubation experiments (Lewicka-
Szczebak et al., 2014, 2015). Thus, we quantified the possible range of variations in the
$\Delta^{17}O$ value of residual $N_2O$ from that of original $N_2O$ to be less than 0.03 ‰ (Figure 7),
based on $\Delta\delta^{18}O$ (less than 10 ‰) and the variation range of $\beta$, from 0.525 to 0.5305.
These possible variations in $\Delta^{17}O$ (less than 0.075 ‰) were much less than the
difference in $\Delta^{17}O$ values between $O_2$ and $NO_2^-$ in the forested soil (0.7 ‰ on average;
Figures 4c). In addition, the possible variation ranges in $\Delta^{17}O$ become much smaller if the
differences in $\beta$ from 0.528 were smaller than those used in the calculations (from 0.525
to 0.5305). Thus, we concluded that the possible variations in the $\Delta^{17}O$ value of $N_2O$
from that of the source molecules of O atoms ($O_2$, $H_2O$, and $NO_2^-$) during the
transformations, including nitrification, denitrification, and reduction, were negligible.

While the $\Delta^{17}O$ values of soil $O_2$ and $H_2O$ used in this study were referred from

atmospheric $O_2$ and rainwater, respectively, the processes in soil, including diffusion and
respiration of $O_2$ and evaporation and infiltration of rainwater, may cause significant
isotopic fractionations of $\delta^{18}O$, which could consequently alter the $\Delta^{17}O$ values of
atmospheric $O_2$ and rainwater. Thus, prior to using $\Delta^{17}O$ values to identify the pathways
of $N_2O$ production in soils, we evaluated the possible variations in the $\Delta^{17}O$ values of $O_2$
and $H_2O$ in soil compared to those of atmospheric $O_2$ and rainwater. The details are
presented below.

For soil $O_2$, Aggarwal and Dillon (1998) measured the $\delta^{18}O$ values in soil gas at a

depth of 3-4 m at a site near Lincoln, Nebraska, USA ranged from +23.3 ‰ to +27.2 ‰,
showing the values were comparable with that of atmospheric $O_2$ (+23.5 ‰ after
adjustment in Aggarwal and Dillon. 1998). This confirms that the isotopic fractionations
of soil $O_2$ induced from soil respiration and diffusion processes weren't significant.
Because the maximum variation in $\delta^{18}O$ from atmospheric $O_2$ to soil $O_2$ was less than
3.7 ‰ (27.2 ‰ − 23.5 ‰), using the method presented in Figure 7, we quantified the
possible variations in the $\Delta^{17}O$ value of soil $O_2$ from that of atmospheric $O_2$ to be less
than 0.01 ‰. Thus, we ignored the negligible variations in this study.

Similarly, for soil $H_2O$, Lyu (2021) observed that $\delta^{18}O$ values in soil $H_2O$ at the depths

of 0-5 cm, 15-20 cm, and 40-45 cm in a subtropical forest plantation ranged from −4 ‰
to −10 ‰, which fully overlapped with local rainwater (−1 ‰ to −16 ‰), indicating
insignificant isotopic fractionations of soil $H_2O$ during hydrological processes such as
infiltration and evaporation. Besides, Aron et al. (2021) compiled $\Delta^{17}O$ values of
terrestrial $H_2O$ including rainwater, surface and subsurface water in earth, ranged from
+0.06 to −0.06 ‰ and didn't show significant difference with each other, which also
indicating that the possible variations of $\Delta^{17}O$ values of soil $H_2O$ compared to that of
rainwater should be negligible. Finally, we added the variations of $\Delta^{17}O$ values (+0.06 to
−0.06 ‰) of terrestrial $H_2O$ reported in Aron et al. (2021) to Figures 4 and 6 as the
uncertainties of $\Delta^{17}O$ values of soil $H_2O$.

In the forested soil, $N_2O$ exhibited $\Delta^{17}O$ values (−0.30±0.09 ‰ on average) that were

close to that of $O_2$ (−0.44 ‰) but deviated from those of soil $NO_2^-$ on fine days
(+0.24±0.14 ‰; Figures 4c and 4d), implying that nitrification was the main pathway to
produce $N_2O$ in the soil on fine days. Conversely, $N_2O$ emitted from the soil on rainy
days exhibited $\Delta^{17}O$ values (+0.12±0.13 ‰) that were close to those of soil $NO_2^-$
(+0.22±0.09 ‰) and soil $H_2O$ (+0.03 ‰) but deviated from that of $O_2$ (Figures 4c and
4d), implying that (1) the main pathway to produce $N_2O$ changed from nitrification on
fine days to denitrification on rainy days and/or (2) the possible contribution of O atoms
derived from soil $H_2O$ became more active during the production of $N_2O$ in the soil on
rainy days.

**4.2 Changes in the $\Delta^{17}O$ of $N_2O$ emitted from artificially fertilized soils**

To quantitatively constrain the possible contributions of O atoms derived from soil

$H_2O$ during the production of $N_2O$ in the soil, we observed changes in the isotopic
compositions of $N_2O$ from the same soil in response to artificial fertilization. In the plot
fertilized with CS, the $\Delta^{17}O$ value of $N_2O$ emitted from the soil ($+7.79\pm0.61$ ‰ on the
average of 2 and 6 days after the fertilization) became significantly closer to that of soil
$NO_2^-$ ($+10.3\pm2.9$ ‰) compared with that of atmospheric $O_2$ ($-0.44$ ‰; Figure 6b). This
suggested that denitrification became the main pathway of $N_2O$ production, probably
because of fertilization, which resulted in a significantly higher concentration of $NO_3^-$
($278.4\pm43.2$ mg N $kg^{-1}$; Table S1) than that of $NH_4^+$ ($15.8\pm4.1$ mg N $kg^{-1}$) in the CS plot.
In addition, $N_2O$ emitted from the CS plot exhibited $\Delta^{17}O$ values that were significantly
different from those of soil $H_2O$ ($+0.03$ ‰; Figure 6b), implying that the contribution of
O atoms derived from soil $H_2O$ was minor during the reduction of $NO_2^-$ to produce $N_2O$.
If all the O atoms with low $\Delta^{17}O$ values in $N_2O$ were derived from soil $H_2O$ ($+0.03$ ‰) in
the CS plot, the contribution of O atoms derived from soil $H_2O$ was calculated to be 24 %
(($10.30$ ‰ – $7.79$ ‰) / ($10.30$ ‰ – $0.03$ ‰)), based on the isotopic mass balance. If the
$O_2$ also contributed to the $N_2O$ production in the CS plot, the contribution of O atoms
derived from soil $H_2O$ should be further reduced. As a result, we determined that the
maximum possible contribution of O atoms derived from soil $H_2O$ during the reduction
of $NO_2^-$ to $N_2O$ was 24 %.

On the other hand, in the plot fertilized with urea (U plot), the $\Delta^{17}O$ value of $N_2O$

($-0.15\pm0.01$ ‰) was close to that of $O_2$ ($-0.44$ ‰) compared with that of soil $NO_2^-$
($+0.24\pm0.10$ ‰). This suggested that nitrification was the main pathway of $N_2O$
production (Figure 6b), probably due to the enhancement of $NH_4^+$ concentration
($423.1\pm18.2$ mg N $kg^{-1}$; Table S1) compared with that of $NO_3^-$ ($13.0\pm10.7$ mg N $kg^{-1}$) in
the U plot. In addition, $N_2O$ emitted from the U plot exhibited $\Delta^{17}O$ values that were
significantly different from that of soil $H_2O$ ($+0.03$ ‰; Figure 6b), implying that the
contribution of O atoms derived from soil $H_2O$ was also minor during the oxidation of
$NH_4^+$ to produce $N_2O$. Consequently, the contribution of O atoms derived from soil $H_2O$
was minor in the soil during $N_2O$ production, irrespective of the pathways of $N_2O$
production being either nitrification or denitrification. In addition, it is difficult to explain
the observed increases in the emission flux of $N_2O$ from the soil on rainy days based only
on the active contribution of O atoms derived from soil $H_2O$. Consequently, we
concluded that $N_2O$ production through denitrification became active in the soil on rainy
days, which resulted in increased $N_2O$ emission and higher $\Delta^{17}O$ values.

**499    4.3 Verification of active $N_2O$ emission by denitrification on rainy days**

The forested soil exhibited significantly lower WFPS on fine days (66.1±6.2 %; Table

S1) than on rainy days (95.6±19.1 %), implying that the $O_2$ concentration in the soil was
higher on fine days than on rainy days. Using the isotope tracer enriched in $^{15}N$ ($^{15}NO_3^-$
or $^{15}NH_4^+$), Mathieu et al. 2006 estimated the relative importance of nitrification and
denitrification to produce $N_2O$ in soil. They found that nitrification produced the majority
of $N_2O$ under low WFPS conditions (75 %), whereas denitrification accounted for more
than 85 % of $N_2O$ produced under high WFPS conditions (150 %). Similarly, using
natural stable isotopes (SP), Ibraim et al. 2019 reported the primary pathway for $N_2O$
production in a grassland shifted from nitrification to denitrification as increasing WFPS,
when WFPS was below 90 %. Thus, we conclude that the lower WFPS in the soil caused
oxic conditions on fine days, resulting in nitrification as the primary pathway for $N_2O$
production in the soil. Conversely, the higher WFPS caused redox conditions in the soil
on rainy days, resulting in active $N_2O$ production through denitrification in the soil
(Figures 4a and 4b).
During continuous monitoring of the emission flux of $N_2O$ from an agricultural soil for
four years, Anthony et al. 2023 found short-term increases in the emission flux during or
immediately after rainfall or irrigation. They referred to this high emission flux as "hot
moments" and defined it as exceeding four standard deviations of that of normal periods.
They also found significant correlations between the emission flux and WFPS, leading to
the conclusion that variations in the concentrations of $O_2$ in surface soils were responsible
for the hot moments of $N_2O$ emissions. Although the hot moments accounted for 1 % of
all measurements, they contributed up to 57 % of the annual emissions, indicating their
significance as a source of atmospheric emissions. In this study, the emission flux of $N_2O$
on rainy days also exceeded four standard deviations of that on fine days (Figures 4a and
4b). The $\Delta^{17}O$ evidence of $N_2O$ found in this study further verified that denitrification
was mainly responsible for the enhancement of $N_2O$ production during the hot moments.

**4.4 Changes in the pathway of $N_2O$ production due to fertilization with urea**
During our observation on the plot fertilized with urea (U plot), $N_2O$ emitted from the
plot exhibited $\Delta^{17}O$ values ($-0.15\pm0.01$ ‰ on average; Figure 6b) that were significantly
higher than those of the plot without fertilization (NF plot; $-0.35\pm0.04$ ‰ on average).
Although an increase in the contribution of O atoms derived from soil $H_2O$ could be
responsible for the $\Delta^{17}O$ values in addition to an increase in $N_2O$ production through
nitrification, we concluded that an increase in $N_2O$ production through $NO_2^-$ reduction
was responsible for the $\Delta^{17}O$ values ($-0.15\pm0.01$ ‰ on average) of $N_2O$ produced in the
plot in response to fertilization of urea/$NH_4^+$ for the following reasons.

Avrahami et al. 2002 reported that fertilization with urea/$NH_4^+$ in soil promoted the

oxidation of $NH_4^+$ and thus provided electron acceptors for denitrification. That is, the
enrichment of nitrate through nitrification also promotes denitrification. Based on the
stable isotopes of $N_2O$ ($\delta^{15}N$, $\delta^{18}O$, and SP), along with in vitro acetylene blockage
experiments on agricultural soils fertilized with $NH_4^+$, Zhang et al. 2016 reported that
while 50 %−70 % of $N_2O$ was produced through nitrification, nitrifier denitrification
($NH_4^+ \rightarrow NO_2^- \rightarrow N_2O$) and/or heterotrophic denitrification ($NH_4^+ \rightarrow NO_3^- \rightarrow$
$NO_2^- \rightarrow N_2O$) accounted for 30 %–50 % of $N_2O$ production. Similar results have also
been reported in previous studies. Although $N_2O$ production through nitrification was
simulated by fertilization with urea/$NH_4^+$ in various soils, denitrification also accounted
for a significant portion of $N_2O$ production (Kaushal et al., 2022; Khalil et al., 2004; Zhu
et al., 2013). In addition to nitrifier/heterotrophic denitrification, $N_2O$ produced through
the anammox process ($NH_4^+ + NO_2^- \rightarrow N_2O$, Okabe et al., 2011; Tang et al., 2011;
Tsushima et al., 2007) can be responsible for the reduction of $NO_2^-$ as well. Zhu et al.
2011 found that the highest rate of anammox was comparable with that of denitrification
in soils fertilized with $NH_4^+$ ($6.2-178.8$ mg N $kg^{-1}$). These previous experiments support
our observation on the U plot that the addition of urea/$NH_4^+$ stimulates $N_2O$ production
through nitrifier denitrification and/or heterotrophic denitrification, and/or anammox
reaction in addition to nitrification. The increased $NO_3^-$ concentration in the U plot
($13.0\pm10.7$ mg N $kg^{-1}$) compared with those in the NF plot ($2.3\pm0.5$ mg N $kg^{-1}$) probably
due to nitrification stimulated by the addition of $NH_4^+$ may be responsible for the active
reduction of $NO_2^-$.

**4.5 Stable $\Delta^{17}O$ as a natural signature for identifying $N_2O$ production pathways**

Although the $\delta^{18}O$ values of $N_2O$ emitted from the soil were significantly higher
than those of the sources of O atoms in $N_2O$ ($NO_2^-$, $O_2$, and $H_2O$; Figures 4e and 6a) due
to the fractionations of oxygen isotopes during the production and/or reduction of $N_2O$,
the $\Delta^{17}O$ values of $N_2O$ remained within the range of these sources. This indicates that
$\Delta^{17}O$ primarily reflects the pathways of $N_2O$ production, providing information distinct
from the $\delta^{18}O$ signature because $\Delta^{17}O$ is stable during the processes of biogeochemical
isotope fractionation. Moreover, while $N_2O$ emission from the forested soil did not show
significant differences in $\delta^{15}N$ and $\delta^{18}O$ values between fine and rainy days due to the
fractionations of nitrogen and oxygen isotopes (Figures 4f and 4h), the significant
difference in the $\Delta^{17}O$ values of $N_2O$ between fine and rainy days (Figure 4d) highlights
$\Delta^{17}O$ to be a promising natural signature for identifying the pathways of $N_2O$ production
in soils.
In addition to natural soils, the stable $\Delta^{17}O$ signature is expected to be useful for
identifying the pathways of $N_2O$ production in various ecosystems, such as agricultural
soils and aquatic environments, where the isotopic fractionations of nitrogen and oxygen
isotopes involving biogeochemical processes are significant as well. However, in order to
identify the pathways of $N_2O$ production quantitatively, the uncertainties, including the $\beta$
values of each reaction during $N_2O$ production and the contributions of O atoms derived
from soil $H_2O$ during $N_2O$ production, should be quantified precisely in the future
studies.

**5. Conclusions**

Temporal variations in $\Delta^{17}O$ of $N_2O$ emitted from forested soil were determined to
identify the main pathway of $N_2O$ production. Both $\Delta^{17}O$ values and fluxes of $N_2O$ were
significantly higher on rainy days compared to fine days. Besides, the $\Delta^{17}O$ values of
$N_2O$ emitted on rainy and fine days were close to those of soil $NO_2^-$ and $O_2$, respectively.
Because $NO_2^-$ and $O_2$ were the source of O-atoms in $N_2O$ production through
denitrification and nitrification, respectively, we concluded that while nitrification
dominated $N_2O$ production on fine days, denitrification became active on rainy days,
resulting in the $N_2O$ flux increasing. In addition, the $\Delta^{17}O$ of $N_2O$ emitted from the same
soil fertilized with either Chile saltpeter or urea exhibited values that were significantly
different from those of soil $H_2O$, implying that the contributions of O atoms derived from
soil $H_2O$ during $N_2O$ production were minor. Furthermore, while $N_2O$ emitted from the
forested soil did not show significant differences in $\delta^{15}N$ and $\delta^{18}O$ values between fine
and rainy days, the significant difference in the $\Delta^{17}O$ values of $N_2O$ highlights $\Delta^{17}O$ to be
a promising natural signature for identifying the pathways of $N_2O$ production in soils,
because $\Delta^{17}O$ is almost stable during isotope fractionation processes such as $N_2O$
production and reduction.

*Data availability.* All the primary data are presented in the Supplement.

*Author contributions.* WD, UT, and FN designed the study. WD, TH, WR, MI, HX, and
YK performed the field observations. WD, UT, TS and FN determined the concentrations
and isotopic compositions of the samples. WD, TS, FN, and UT performed data analysis.

*Competing interests.* The authors declare that they have no conflict of interest.

*Acknowledgments.*
We thank the anonymous referees for their valuable remarks on an earlier version of
this paper. We are grateful to the members of the Biogeochemistry Group at Nagoya
University for their valuable support throughout this study. This work was supported by a
Grant-in-Aid for Scientific Research from the Ministry of Education, Culture, Sports,
Science, and Technology of Japan under grant numbers 22H00561, 17H00780,
22K19846, the Grant-in-Aid for Japan Society for the Promotion of Science Fellows
under grant number 23KJ1088, the Yanmar Environmental Sustainability Support
Association, the River Fund of the River Foundation, Japan, the Reiwa Environmental
Foundation, and the National Research Foundation of Korea Grant from the Korean
Government (MSIT; the Ministry of Science and ICT, NRF-2021M1A5A1065425,
KOPRI-PN24011).

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

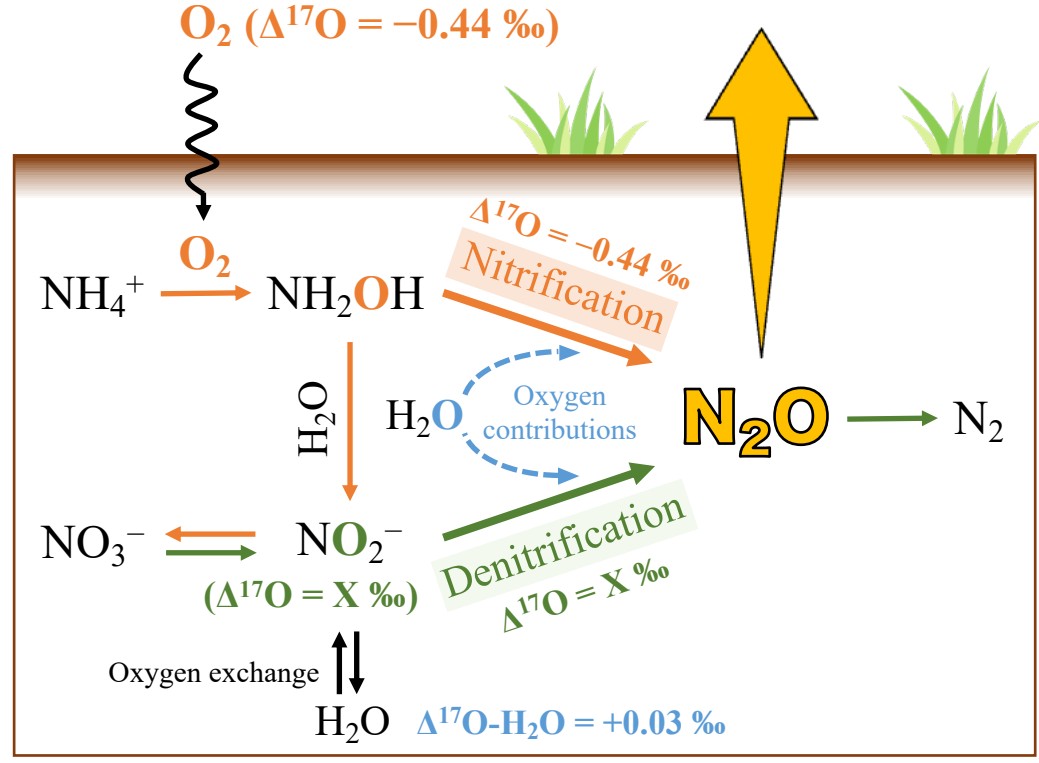

**Figure 1.** Schematic showing the pathways of $N_2O$ production in soil (Kool et al., 2007,
2011; Wankel et al., 2017; Wrage et al., 2005) and the $\Delta^{17}O$ values of $O_2$ (Sharp et al.,
2016), $NO_2^-$, and $H_2O$ (Uechi and Uemura, 2019). The orange lines, green lines, and blue
dash lines indicate the processes of nitrification, denitrification, and the possible
contributions of O atoms derived from soil $H_2O$ through nitrification and denitrification,
respectively.

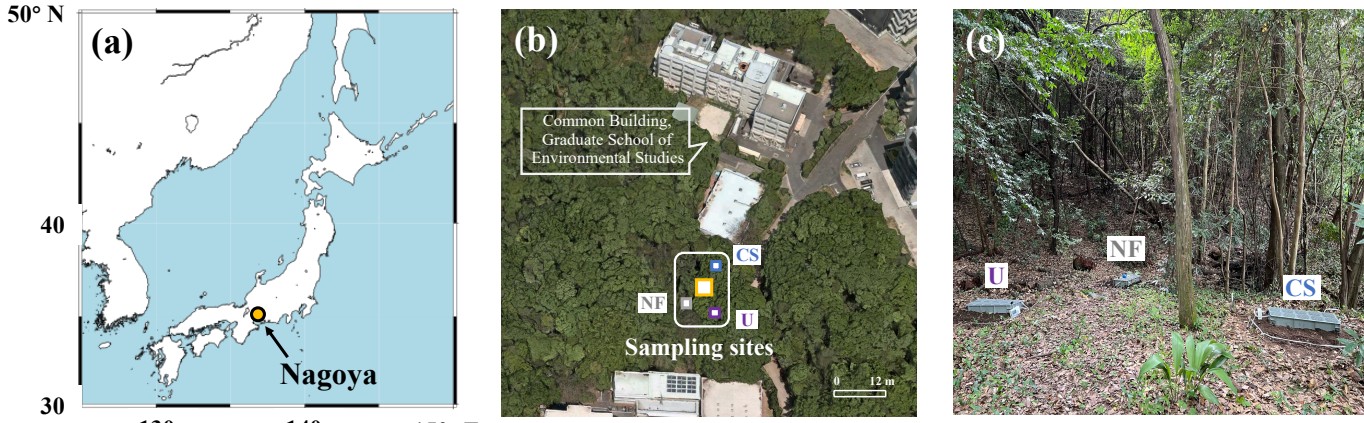

**Figure 2.** Map showing the location of Nagoya, Japan, where the studied site is located
(a). Map showing the monitoring site of $N_2O$ emitted from forested soil in a secondary
warm-temperate forest (yellow square) and the plots fertilized with Chile saltpeter (CS,
blue square), urea (U, purple square), and no fertilizer (NF, gray square) (b). Photo
showing the plots and flow chambers set on the plots (c).

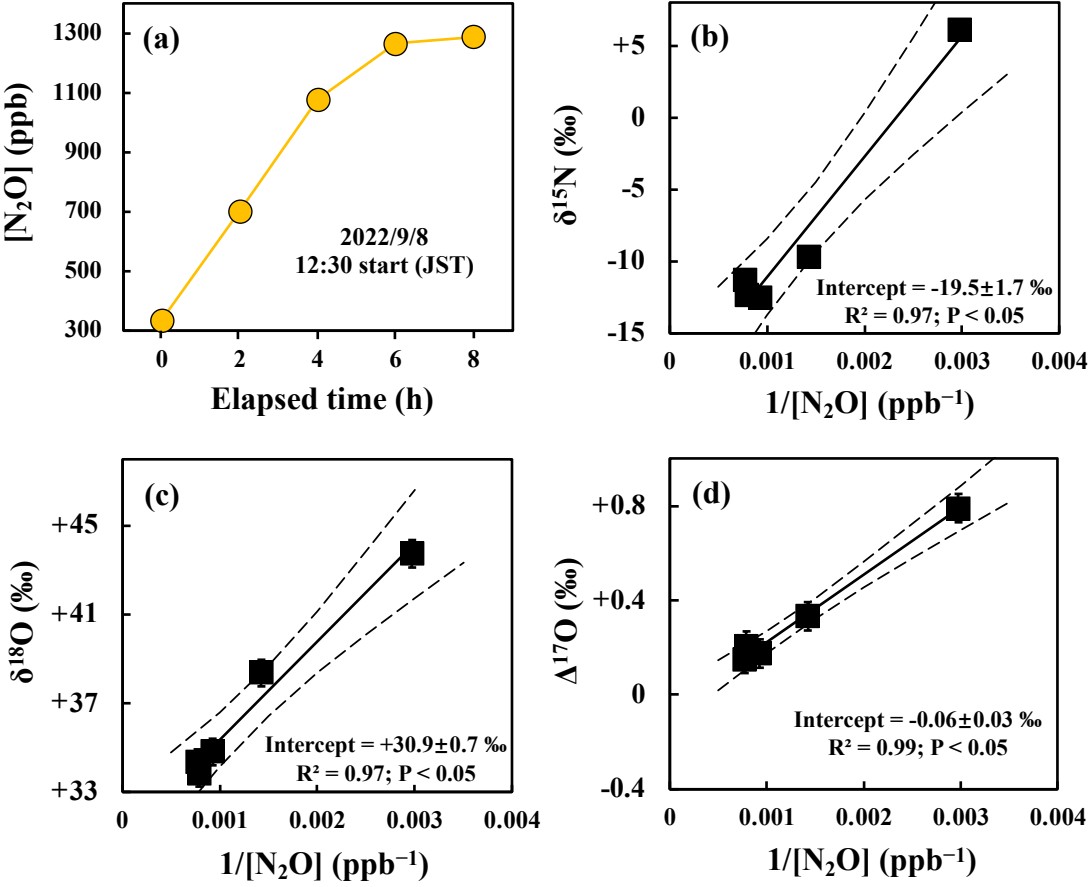

**Figure 3.** An example of changes in the concentration of $N_2O$ ($[N_2O]$) in gas samples during the observation on September 8, 2022, plotted as a function of the elapsed time since the deployment of the flow chamber on the forested soil (a), and the $\delta^{15}N$ (b), $\delta^{18}O$ (c), and $\Delta^{17}O$ (d) values of $N_2O$ plotted as a function of the reciprocal of $[N_2O]$ ($1/[N_2O]$) during the observation. Each solid line is the least squares fitting of the samples, while each dotted line is the $2\sigma$ confidence interval of the fitting line. Error bars smaller than the sizes of the symbols are not shown.

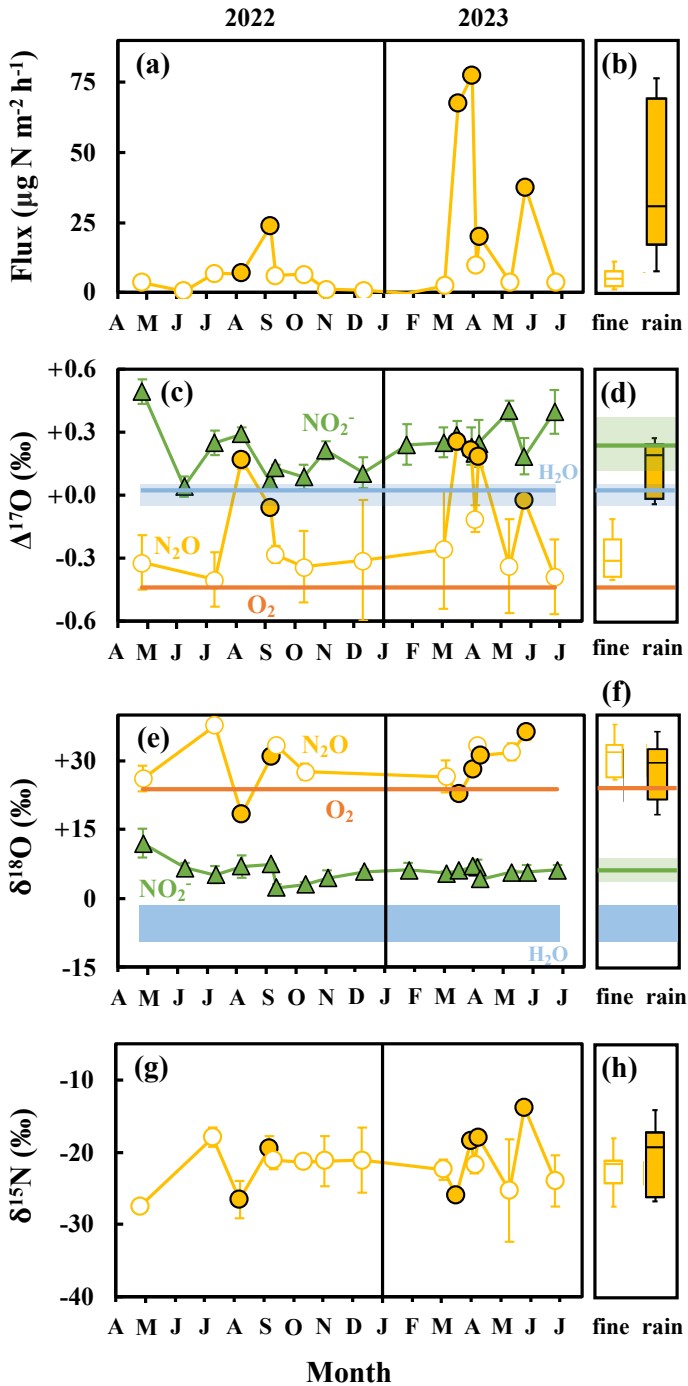

Figure 4. Temporal variations in the flux (a), $\Delta^{17}O$ (c), $\delta^{18}O$ (e), and $\delta^{15}N$ (g) values of

N$_2$O emitted from the forested soil, and the $\delta^{18}O$ and $\Delta^{17}O$ values of soil NO$_2^-$ (green

triangles), O$_2$ (orange lines), and soil H$_2$O (blue area or line). Sampling performed on fine

and rainy days is indicated by the open (white) and solid (yellow) circles, respectively,

with the box plots of the emission flux (b), $\Delta^{17}O$ (d), $\delta^{18}O$ (f), and $\delta^{15}N$ (h) of $N_2O$ on

fine and rainy days. The black lines of the box plots indicate the median values. The

lower and upper boundaries of the box plots indicate the lower (25 %) and upper (75 %)

quartiles of data for each component, respectively. The whiskers of the box plots denote

the entire range of values for each component. Error bars smaller than the sizes of the

symbols are not shown.

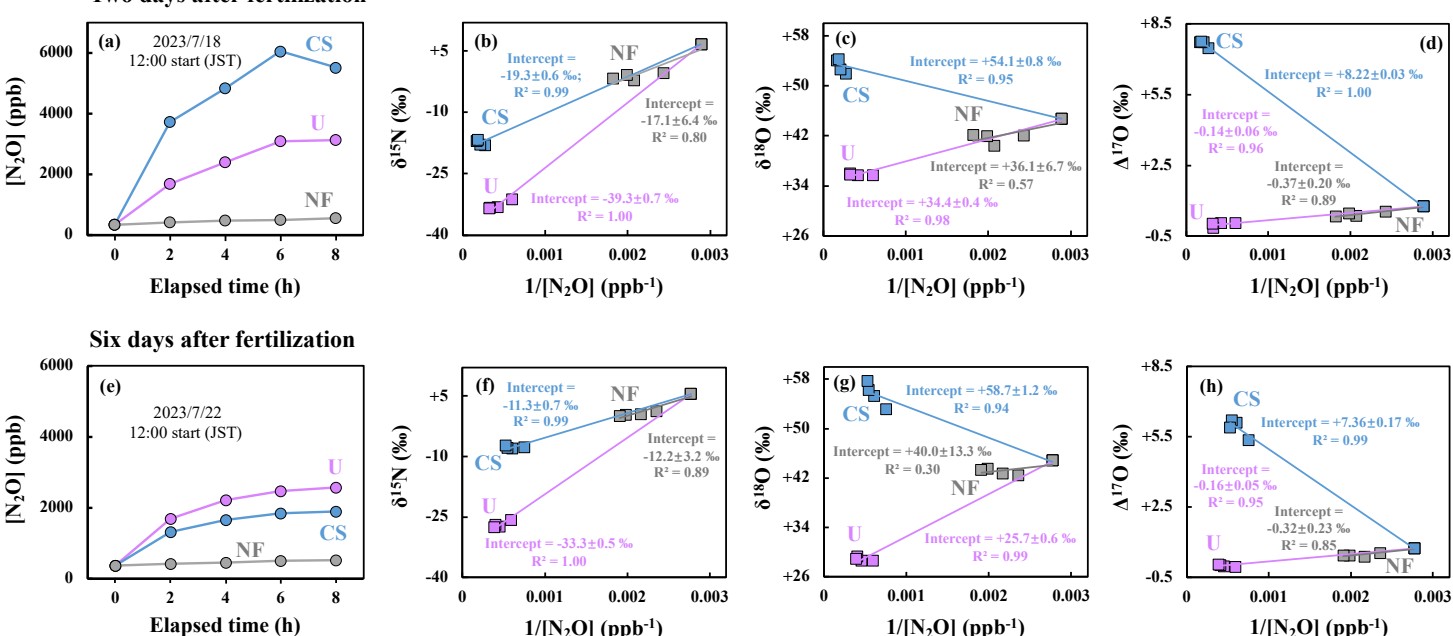

**Figure 5.** Changes in $[N_2O]$ of gas samples collected from the plots of NF (gray), U

(purple), and CS (blue) 2 days after fertilization (a) and 6 days after fertilization (e) and

plotted as a function of the elapsed time since the deployment of the flow chamber; the

$\delta^{15}N$ (b and f), $\delta^{18}O$ (c and g), and $\Delta^{17}O$ (d and h) values of $N_2O$ plotted as a function of

the reciprocal of $[N_2O]$ ($1/[N_2O]$). Error bars smaller than the sizes of the symbols are not

shown.

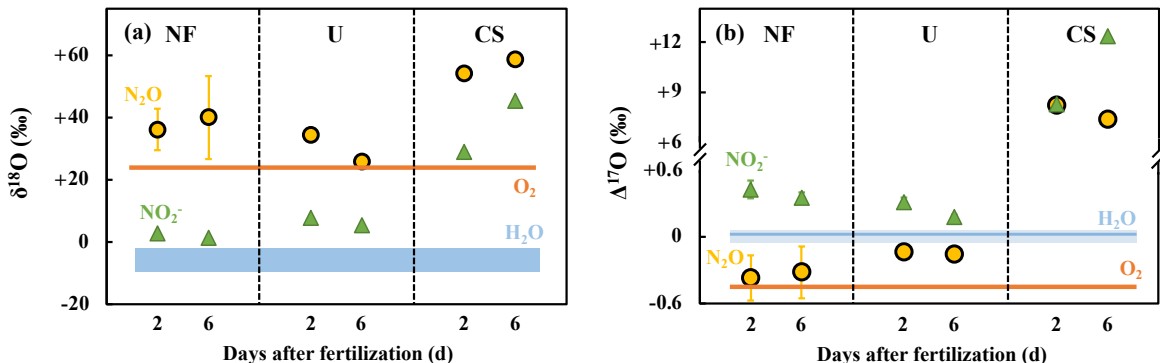

**Figure 6.** The $\delta^{18}O$ (a) and $\Delta^{17}O$ (b) values of $N_2O$ (yellow circles) and $NO_2^-$ (green
triangles) in NF, U, and CS plots determined 2 and 6 days after fertilization, and the $\delta^{18}O$
and $\Delta^{17}O$ values of $O_2$ (orange lines) and soil $H_2O$ (blue area or line). Error bars smaller
than the sizes of the symbols are not shown.

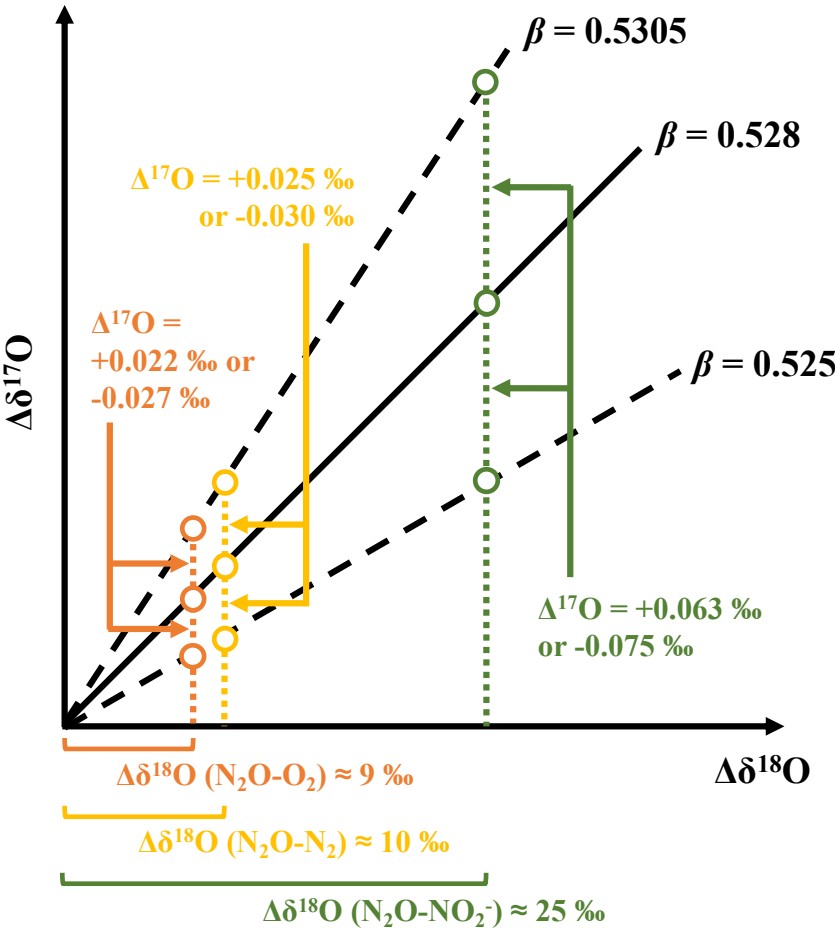

Figure 7. Schematic showing the possible variations in the $\Delta^{17}O$ value of $N_2O$ from that of the source of O atoms ($O_2$ and $NO_2^-$) during transformations, including nitrification (orange circles), denitrification (green circles), and reduction (yellow circles), due to variations in isotope fractionation and $\beta$ from 0.525 to 0.5305.