# Peer review of "Triple oxygen isotope evidence for the pathway of nitrous oxide"

_EGUsphere, 2025_

## Author Comment (AC1)

Dear Referee #1

Thank you very much for your valuable comments on our manuscript. We would like to respond to each of your comments one by one.

**[1] Section 2.3: While extraction with 2 M KCl is standard for measuring extractable soil nitrate and ammonium, significant loss of soil nitrite can occur during the extraction. This issue is well recognized in the soil nitrogen cycling community (e.g., Homyak et al., 2015). Given that nitrite concentration and $\Delta^{17}O$ measurements are critical to this study, could the authors discuss how potential nitrite loss and associated isotopic fractionation might impact their analysis?**

Thank you for your question. While the 2M KCl extraction is widely used for soil nitrite ($NO_2^-$) analysis (Lewicka-Szczebak et al., 2021; Shen et al., 2003), we also notice the concerns raised by Homyak et al. (2015) regarding possible underestimation of soil nitrite concentrations when using KCl solutions compared to deionized water.

To evaluate this potential issue, we conducted a comparative experiment in April 2022 prior to this study. We collected a soil sample from our study site (secondary warm-temperate forest, Figure 2b), which was thoroughly homogenized and divided into two 50 g subsamples. Each subsample was then extracted with either 50 mL of 2M KCl solution or 50 mL MQ water, following the same analytical procedures described in Sections 2.3 and 2.7 of the manuscript.

Our results showed consistent values between the two extraction methods: the KCl-extracted sample yielded a nitrite concentration of 0.90 μM with $\Delta^{17}O$ of 0.55±0.1‰, while the MQ water-extracted sample showed 0.98 μM nitrite with $\Delta^{17}O$ of 0.62±0.1‰. Because both the concentration and $\Delta^{17}O$ value of soil nitrite in KCl solution and MQ water showed no significant differences, we concluded that for our soil type and experimental conditions, the use of 2M KCl solution introduced negligible bias in terms of nitrite recovery or $\Delta^{17}O$ measurements compared to MQ water extraction.

**[2] Section 2.4: The manuscript uses a β (triple oxygen proportionality factor) range of 0.525 to 0.5305 to quantify the potential impact on $\Delta^{17}O$. Although several references are cited, it is unclear why this specific range was chosen. Please clarify. In particular, earlier studies (e.g., Matsuhisa et al., 1978; summarized by Miller, 2002) reported lower β values (e.g., ~0.5164). How would using lower β values affect the results?**

Thank you for the comment. Our selection of this range was based on evidence from recent experimental and theoretical studies examining oxygen isotope fractionation across various compounds (CO, $O_2$, NO, $CO_2$, $NO_2$, $H_2O$, $SO_2$, $SO_3$,

$CO_3^{2-}$, and $SiO_2$), as documented by (Cao and Liu, 2011; Pack and Herwartz, 2014; Sharp and Wostbrock, 2021). These studies collectively demonstrate that this range encompasses most equilibrium and kinetic fractionation processes. Thus, for the calculation of $\Delta^{17}O$ value of $N_2O$, we adopted the midpoint value ($\beta = 0.528$) of this range.

Regarding your concern about lower $\beta$ values reported in earlier work (Matsuhisa et al., 1978), we note that while Matsuhisa et al. (1978) did observe $\beta$ values as low as 0.5164 in some terrestrial rock and water samples, they ultimately choose 0.52 for the quartz-water system as the most representative value. To address how such lower $\beta$ values might affect our results, we quantified the possible variations in the $\Delta^{17}O$ values of $N_2O$ during each reaction using $\beta = 0.52$. Our calculations (following the methodology in Section 4.1 and Figure 7 in the manuscript) show that this would introduce variations in $\Delta^{17}O$ values of $N_2O$ (derived from soil $NO_2^-$ and $O_2$) of less than 0.2 ‰ (Figure R1). This potential variation is significantly smaller than the observed $\Delta^{17}O$ difference between $O_2$ and $NO_2^-$ in our forested soil samples (average 0.7‰; Figure 4c). We concluded that even using such low $\beta$ value ($\beta = 0.52$), our key findings or interpretations can't be affected.

[Figure]

**Figure R1.** Schematic showing the possible variations in the $\Delta^{17}O$ value of $N_2O$ from that of the source of O atoms ($O_2$ and $NO_2^-$) during transformations, including nitrification (orange circles), denitrification (green circles), and reduction (yellow circles), due to variations in isotope fractionation and $\beta$ from 0.520 to 0.5305

**[3] Lines 362–363 and throughout the analysis: A constant $\Delta^{17}O$ value was assumed for $O_2$ (-0.44‰) and soil $H_2O$ (0.03‰). Please clarify whether these values could vary due to hydrological and biogeochemical cycling. For instance, could $O_2$ diffusion and heterotrophic consumption affect $O_2$ $\Delta^{17}O$, or could evaporation significantly alter soil $H_2O$ $\Delta^{17}O$?**

Thank you for the question. The $\Delta^{17}O$ values of $O_2$ (−0.44 ‰) and soil $H_2O$ (+0.03 ‰) used in this study was referred from that of atmospheric $O_2$ (Sharp et al., 2016) and rainwater (Uechi and Uemura, 2019), respectively.

For soil $O_2$, Aggarwal and Dillon (1998) measured $\delta^{18}O$ values in soil gas at a depth of 3-4 m at a site near Lincoln, Nebraska, USA ranged from +23.3 ‰ to +27.2 ‰ (Table R1), showing the values were comparable with that of atmospheric $O_2$ (+23.5 ‰ after adjustment in Aggarwal and Dillon. 1998). This confirms that the isotopic fractionation of soil $O_2$ induced from soil respiration and diffusion processes wasn't significant. Because the maximum variation in $\delta^{18}O$ from atmospheric $O_2$ to soil $O_2$ was less than 1.9 ‰ (27.2 ‰ − 23.5 ‰), using the method presented in Section 4.1 and Figure 7, we quantified the possible variations in the $\Delta^{17}O$ value of soil $O_2$ from that of atmospheric $O_2$ to be less than 0.006 ‰. Thus, we ignored the negligible variations in the manuscript.

Similarly, for soil $H_2O$, Lyu (2021) observed that $\delta^{18}O$ values in soil $H_2O$ at the depths of 0-5 cm, 15-20 cm, and 40-45 cm in a subtropical forest plantation ranged from −4 ‰ to −10 ‰ (Figure R2), which fully overlapped with local rainwater (−1 ‰ to −16 ‰), indicating insignificant isotopic fractionations of soil $H_2O$ during hydrological process such as infiltration and evaporation compared to rainwater. Besides, Aron et al. (2021) compiled $\Delta^{17}O$ values of terrestrial $H_2O$ including rainwater, surface and subsurface water in earth, ranged from +0.06 to −0.06 ‰ and didn't show significant difference with each other, which also indicating that the possible variations of $\Delta^{17}O$ values of soil $H_2O$ compared to that of rainwater should be negligible. Finally, we would like to add the variations of $\Delta^{17}O$ values (+0.06 / −0.06 ‰) of terrestrial $H_2O$ reported in Aron et al. (2021) to Figures 4 and 6 as the uncertainties of $\Delta^{17}O$ values of soil $H_2O$ in the revised manuscript.

**Table R1.** Concentration and isotopic compositions of soil gas oxygen and carbon dioxide from the midwestern USA site (Aggarwal and Dillon. 1998).

| Location | Depth (m) | Date | $CO_2$ (%) | $O_2$ (%) | $\delta^{18}O\text{-}O_2$ [a] ‰, SMOW | $\delta^{13}C\text{-}CO_2$ [a] ‰, PDB |
|---|---|---|---|---|---|---|
| 1b | 2.9 | June 94 | 1.8 | 13.8 | 27.2 | −21.8 |
|  | 2.9 | Dec. 94 | 2.3 | 15.1 | 24.2 | −21.7 |
| 2b | 3.4 | June 94 | 0.7 | 16.3 | 25.1 | −22.4 |
| 2b | 3.4 | Dec. 94 | 0.9 | 15.2 | 23.3 | −22.8 |
| 3b | 4.1 | June 94 | 0.7 | 15.7 | 25.3 | −22.1 |
| 3b | 4.1 | Dec. 94 | 1.4 | 15.2 | 25.1 | −24.0 |
| 5b | 3.2 | June 94 | 0.8 | 17.6 | 24.7 | −21.7 |
| 5b | 3.2 | Dec. 94 | 1.0 | 16.3 | 24.7 | −21.0 |
| 4b | 3.2 | June 94 | 2.3 | 17.1 | 24.5 | −20.0 |
| 4b | 3.2 | Dec. 94 | 3.0 | 16.0 | 24.0 | −19.9 |

[a] Isotopic values reported are averages of duplicate analyses with a standard deviation of 0.3‰ for $\delta^{18}O$ and 0.2‰ for $\delta^{13}C$. The oxygen isotope ratios have been adjusted to an atmospheric oxygen $\delta^{18}O$ of 23.5 ‰.

[Figure]

**Figure R2.** Temporal variations of the amount of precipitation, $\delta^{18}O$ in precipitation, and weighted average $\delta^{18}O$ in soil water source during winter, spring, summer, and autumn, at 0–5 (0–5 cm), 15–20 (15–20 cm), and 40–45 (40–45 cm) cm depths (Lyu. 2021).

**[4] Lines 378–397: The characterization of $\delta^{18}O$ offsets between $O_2$ and $N_2O$, and between $NO_2^-$ and $N_2O$, does not necessarily represent true isotope effects between $N_2O$ and its oxygen precursors because field-measured $N_2O$ is a mixture of multiple sources. For example, in Fig. 6a, the actual $\delta^{18}O$ difference between $NO_2^-$ and $N_2O$ may be larger than calculated if $O_2$-derived $N_2O$ has $\delta^{18}O$ values similar to that of $O_2$. Similarly, the $O_2$-$N_2O$ difference may be smaller than estimated. This mixing effect could confound the use of $\delta^{18}O$ differences to estimate $\Delta^{17}O$ variations and warrants further clarification.**

Thank you for your comment. You mentioned that the true field-measured $N_2O$ is a mixture of multiple sources is correct. However, the mixture ratios of $N_2O$ produced through nitrification and denitrification were unknown. Thus, in our theoretical calculations for the possible variations in the $\Delta^{17}O$ values of $N_2O$ in Section 4.2, we separated the nitrification and denitrification to discuss the possible variations.

After we supposed that if all O atoms in $N_2O$ were derived from $O_2$ (Lines 377-378), the average in $\delta^{18}O$ from $O_2$ to $N_2O$ due to nitrification ($\Delta\delta^{18}O$ ($N_2O-O_2$)) was estimated to be 9 ‰ on average. Similarly, after we supposed that if all O atoms in $N_2O$ were derived from $NO_2^-$ (Lines 391-392), the average variation in $\delta^{18}O$ from $NO_2^-$ to $N_2O$ due to fractionation ($\Delta\delta^{18}O(N_2O-NO_2^-)$) was estimated to be 25 ‰ on average.

**[5] Fig. 7 and related discussion: I commend the authors for conducting a sensitivity analysis to assess how much $\Delta^{17}O$ variation may stem from biogeochemical processes versus purely geochemical processes (i.e., $\beta$ variability). However, applying the $\beta$ range to the net $\delta^{18}O$ difference between $N_2O$ and oxygen sources treats the $N_2O$-producing processes as a single step. In reality, processes like nitrite reduction involve multiple sub-steps (e.g., $NO_2^-$ to NO, NO to $N_2O$, isotope exchange with $H_2O$), each potentially associated with different $\beta$**

**values. This could lead to larger $\Delta^{17}O$ variations than those estimated from a single-step approach. This limitation should be discussed.**

Thank you for your comment. Because the $\beta$ values for processes of $NO_2^-$ to NO and NO to $N_2O$ should be included in the range of 0.525 to 0.5305 (Cao and Liu, 2011; Matsuhisa et al., 1978; Pack and Herwartz, 2014; Sharp and Wostbrock, 2021), the processes of $NO_2^-$ to NO and NO to $N_2O$ were merged into the process of denitrification for the theoretical calculations for the possible variations in the $\Delta^{17}O$ values of $N_2O$. As a result, the calculated possible variations in $\Delta^{17}O$ during denitrification (less than 0.075 ‰) incorporated the $\beta$ variability across these sub-steps.

Besides, because the estimated $\Delta^{17}O$ values of soil $NO_2^-$ included the effect of oxygen isotope exchange between soil $NO_2^-$ and $H_2O$, the oxygen isotope exchange between soil $NO_2^-$ and $H_2O$ can't affect the $\Delta^{17}O$ values of $N_2O$.

Additionally, because we concluded the contributions of O atoms derived from soil $H_2O$ were minor during the reduction of $NO_2^-$ and oxidation of $NH_4^+$ to produce $N_2O$, the discussion for possible variations in $\Delta^{17}O$ values of $N_2O$ due to the process of the contribution of O atoms derived from soil $H_2O$ was ignored.

**[6] Lines 434–438: It is unclear how the 24% contribution of soil $H_2O$ was derived.**

This calculation was based on isotopic mass balance. In the plot fertilized with CS, the average $\Delta^{17}O$ value of $N_2O$ emitted from the soil 2 and 6 days after the fertilization was +7.79 ‰ (Lines 429-431). The $\Delta^{17}O$ value of the possible source of O atoms in $N_2O$ was +10.30 ‰ for soil $NO_2^-$, +0.03 ‰ for soil $H_2O$, and −0.44 ‰ for $O_2$, respectively. If all O atom in $N_2O$ were derived from soil $H_2O$ (+0.03 ‰), the contribution of O atoms derived from soil $H_2O$ was calculated to be 24 % ((10.30 ‰ – 7.79 ‰) / (10.3 ‰ – 0.03 ‰)). If the $O_2$ also contributes to the $N_2O$ production, the contribution of O atoms derived from soil $H_2O$ should be smaller (less than 24 %). As a result, we concluded that the maximum possible contribution of O atoms derived from soil $H_2O$ during the reduction of $NO_2^-$ to $N_2O$ was 24 % (Lines 428-439). We would like to clarify that in the revised manuscript.

**Additionally, Fig. 6b shows that the $\Delta^{17}O$ of $N_2O$ in the CS plot was significantly lower than that of $NO_2^-$ six days after tracer addition. This suggests that soil $H_2O$ may have played a significant role during nitrite reduction to $N_2O$.**

Thank you for your comment. Compared to the value observed 2 days after fertilization, the $\Delta^{17}O$ value of $N_2O$ emitted from the soil in the CS plot 6 days after fertilization became lower than that of soil $NO_2^-$ (Figure 6b), implying that (1) the soil $H_2O$ have played a significant role 6 days after fertilization as suggested, or (2) the

relative contribution of nitrification to $N_2O$ production increased 6 days after fertilization. Because the main pathway to produce $N_2O$ was nitrification in the NF plot (no fertilizer addition) (Figure 6b), the diminishing fertilization effect over time resulted in reduced $N_2O$ production through denitrification was responsible for the relative contribution of nitrification to $N_2O$ production increased 6 days after fertilization. The significant decrease in $N_2O$ flux from 112.3 to 39.4 $\mu g\,N\,m^{-2}\,h^{-1}$ between 2 and 6 days after fertilization further confirm the diminishing fertilization effect over time.

Importantly, similar to 2 days after fertilization, the $\Delta^{17}O$ value of $N_2O$ emitted from the soil in the CS plot 6 days after fertilization (+7.36 ‰) remained closer to that of soil $NO_2^-$ (+12.32 ‰) than that of atmospheric $O_2$ (−0.44 ‰) and $H_2O$ (+0.03 ‰), consistent with our conclusion (Lines 428-432) that the denitrification became the main pathway of $N_2O$ production in the CS plot.

**[7] Lines 444–449 and Fig. 6b: Apparent differences in $\Delta^{17}O$ between soil $H_2O$ and $N_2O$ cannot be used to conclusively rule out $H_2O$ contributions during $N_2O$ production. In the NF and U plots, the $\Delta^{17}O$ of soil $H_2O$ lies between that of $NO_2^-$ and $N_2O$, and both soil $H_2O$ and $NO_2^-$ have higher $\Delta^{17}O$ than $O_2$. Could significant $H_2O$ exchange during $N_2O$ production explain these observations, leading to a mixed $\Delta^{17}O$ signal from both $H_2O$- and $O_2$-derived $N_2O$?**

Thank you for your comment.

In NF plot, the average $\Delta^{17}O$ value of $N_2O$ (−0.35 ‰) measured 2 and 6 days after fertilization was close to that of $O_2$ (−0.44 ‰) compared to that of soil $H_2O$ (+0.03 ‰) and soil $NO_2^-$ (+0.38 ‰) (Figure 6b), implying that the O atoms in $N_2O$ mainly derived from $O_2$ rather than soil $H_2O$. Thus, the $H_2O$ contribution during $N_2O$ production can't be significant in this case.

In U plot, while the significant $H_2O$ contribution during $N_2O$ production could explain the $\Delta^{17}O$ value of $N_2O$ becoming higher than that in NF plot after fertilization, the observed increases in the emission flux of $N_2O$ from the soil in NF plot (from 4.7 to 63.7 $\mu g\,N\,m^{-2}\,h^{-1}$; Table S1 in supplement) can't be explained by the significant $H_2O$ contribution during $N_2O$ production. Thus, we maintain our conclusion that the increase in $N_2O$ production through $NO_2^-$ reduction was responsible for the $\Delta^{17}O$ values of $N_2O$ produced in the U plot in response to fertilization of urea/$NH_4^+$ for the reasons described in the manuscript (Lines 490-514).

**[8] Section 4.5: Early in the manuscript, the authors argue that bulk isotopic and SP-based techniques for $N_2O$ source apportionment are limited due to isotopic fractionations during cycling (lines 56-61), whereas $\Delta^{17}O$ measurements may be more robust. After presenting the results, I would encourage the authors to revisit this point with more specificity. Given potential complications such as $H_2O$ exchange and multiple contributing sources ($H_2O$, $O_2$, $NO_2^-$), can $\Delta^{17}O$**

**measurements realistically achieve quantitative source apportionment? If so, what would the total uncertainty be, considering analytical precision, β variability, and uncertainties from the Keeling approach? Under what conditions would Δ¹⁷O approaches be preferable to conventional methods, and when might they be less effective?**

Thank you for your suggestion. We would like to add the uncertainty information for using $\Delta^{17}O$ as a natural signature for identifying $N_2O$ production pathways including $H_2O$ contributions, analytical precision, and β variability in Section 4.5 in the revised manuscript.

We would like to thank you for the helpful comments. We hope that our responses to your comments are satisfactory.

Sincerely,
Weitian Ding
Graduate School of Environmental Studies,
Nagoya University
Furo-cho, Chikusa-ku, Nagoya,
464-8601, JAPAN
E-mail: dwt530754556@gmail.com
Cc: Drs. Urumu Tsunogai and Fumiko Nakagawa

**Reference**

Aggarwal, P. K. and Dillon, M. A.: Stable Isotope Composition of Molecular Oxygen in Soil Gas and Groundwater: A Potentially Robust Tracer for Diffusion and Oxygen Consumption Processes, Geochimica et Cosmochimica Acta, 62, 577–584, https://doi.org/10.1016/S0016-7037(97)00377-3, 1998.

Aron, P. G., Levin, N. E., Beverly, E. J., Huth, T. E., Passey, B. H., Pelletier, E. M., Poulsen, C. J., Winkelstern, I. Z., and Yarian, D. A.: Triple oxygen isotopes in the water cycle, Chemical Geology, 565, 120026, https://doi.org/10.1016/j.chemgeo.2020.120026, 2021.

Cao, X. and Liu, Y.: Equilibrium mass-dependent fractionation relationships for triple oxygen isotopes, Geochimica et Cosmochimica Acta, 75, 7435–7445, https://doi.org/10.1016/j.gca.2011.09.048, 2011.

Homyak, P. M., Vasquez, K. T., Sickman, J. O., Parker, D. R., and Schimel, J. P.: Improving Nitrite Analysis in Soils: Drawbacks of the Conventional 2 M KCl Extraction, Soil Science Society of America Journal, 79, 1237–1242, https://doi.org/10.2136/sssaj2015.02.0061n, 2015.

Lewicka-Szczebak, D., Jansen-Willems, A., Müller, C., Dyckmans, J., and Well, R.: Nitrite isotope characteristics and associated soil N transformations, Sci Rep, 11, 5008, https://doi.org/10.1038/s41598-021-83786-w, 2021.

Lyu, S.: Variability of $\delta^2H$ and $\delta^{18}O$ in Soil Water and Its Linkage to Precipitation in an East Asian Monsoon Subtropical Forest Plantation, Water, 13, 2930, https://doi.org/10.3390/w13202930, 2021.

Matsuhisa, Y., Goldsmith, J. R., and Clayton, R. N.: Mechanisms of hydrothermal crystallization of quartz at 250°C and 15 kbar, Geochimica et Cosmochimica Acta, 42, 173–182, https://doi.org/10.1016/0016-7037(78)90130-8, 1978.

Pack, A. and Herwartz, D.: The triple oxygen isotope composition of the Earth mantle and understanding $\Delta^{17}O$ variations in terrestrial rocks and minerals, Earth and Planetary Science Letters, 390, 138–145, https://doi.org/10.1016/j.epsl.2014.01.017, 2014.

Sharp, Z. D. and Wostbrock, J. A. G.: Standardization for the Triple Oxygen Isotope System: Waters, Silicates, Carbonates, Air, and Sulfates, Reviews in Mineralogy and Geochemistry, 86, 179–196, https://doi.org/10.2138/rmg.2021.86.05, 2021.

Sharp, Z. D., Gibbons, J. A., Maltsev, O., Atudorei, V., Pack, A., Sengupta, S., Shock, E. L., and Knauth, L. P.: A calibration of the triple oxygen isotope fractionation in the $SiO_2$–$H_2O$ system and applications to natural samples, Geochimica et Cosmochimica Acta, 186, 105–119, https://doi.org/10.1016/j.gca.2016.04.047, 2016.

Shen, Q. R., Ran, W., and Cao, Z. H.: Mechanisms of nitrite accumulation occurring in soil nitrification, Chemosphere, 50, 747–753, https://doi.org/10.1016/S0045-6535(02)00215-1, 2003.

Uechi, Y. and Uemura, R.: Dominant influence of the humidity in the moisture source region on the $^{17}O$-excess in precipitation on a subtropical island, Earth and Planetary Science Letters, 513, 20–28, https://doi.org/10.1016/j.epsl.2019.02.012, 2019.

---

## Author Comment (AC2)

Dear Referee #2

Thank you very much for your valuable comments on our manuscript. We would like to respond to each of your comments one by one.

Major comments:
**1. The study site is very local. The study site's soil type, vegetation, and climate should be contextualized relative to other ecosystems to assess broader applicability.**

Thank you for your comment. In this study, we focused on monitoring the temporal variations of $\Delta^{17}O$ of soil $N_2O$ to evaluate whether the $\Delta^{17}O$ of $N_2O$ can be a signature for identifying the main pathway of $N_2O$ production.

While the current work emphasizes temporal variations of $\Delta^{17}O$ of soil $N_2O$ at a forested soil, we acknowledge the importance of contextualizing these findings across diverse ecosystems. In futural studies, we plan to investigate the spatial variations of $\Delta^{17}O$ of soil $N_2O$ to access the broader applicability and the connections with the variations of soil type, vegetation, and climate.

**In addition, the sample numbers of N2O gas samples are very limited in this study. For instance, only five data and 18 data are illustrated in Figure 3 and 4.**

In Figure 3, the 5 data points represent an example of changes in the concentration and isotopic compositions ($\delta^{15}N$, $\delta^{18}O$, and $\Delta^{17}O$) of $N_2O$ in gas samples during the observation on September 8, 2022. To maintain conciseness, we presented only a subset of data (5 data) in Figure 3 in the main text, while the complete dataset (89 data points) is provided in the supplement (Figure S5).

In Figure 4, 18 data points were estimated from the complete dataset (89 data) using the Keeling plot approach. Importantly, these 18 data points fully support our key findings: (1) $N_2O$ emitted from the soil exhibited significantly higher $\Delta^{17}O$ values on rainy days (+0.12±0.13 ‰) than on fine days (−0.30±0.09 ‰) and (2) the emission flux of $N_2O$ was significantly higher on rainy days (38.8±28.0 μg N m$^{-2}$ h$^{-1}$) than on fine days (3.8±3.1 μg N m$^{-2}$ h$^{-1}$).

**The confidence degree of the line fitting and the representative of the results should be further clarified.**

We would like to add the confidence degree of the line fitting in Figure 3 in the revised manuscript. We would also like to emphasize that Figure 3 represent an example of changes in the concentration and isotopic compositions of $N_2O$ in gas samples during the observation on September 8, 2022.

**The novelty of the findings and this study should be further highlighted by comparing with prior soil Δ¹⁷O studies.**

Thank you for your advice. We would like to further highlight the novelty of this study by comparing it with the prior soil $\Delta^{17}O$ study in the revised manuscript as follows.

While Komatsu et al. (2008) first estimated the $\Delta^{17}O$ of $N_2O$ emitted from a soil to assess whether soil $N_2O$ could be the source of atmospheric $N_2O$'s high $\Delta^{17}O$ ($\Delta^{17}O = + 0.7$ ‰), our study provides the first report of temporal variations of the $\Delta^{17}O$ values of soil $N_2O$. Furthermore, we first propose that $\Delta^{17}O$ can serve as a natural signature for identifying the main $N_2O$ production pathways.

**2. The sample information is missing. The definition of "fine days" and "rainy days" (e.g., precipitation threshold, duration) must be clarified to ensure reproducibility.**

In the manuscript, we have already defined the fine days and rainy days as follows. A fine day is defined as a day without precipitation for 48 hours prior to the end of each sampling. The total precipitation within 12 h at the end of each sampling of the rainy days exceeded 12 mm (Lines 114-116).

**In addition to weather conditions (fine or rainy), the other influencing factors are not considered and discussed, for instance, soil physical, chemical and microbiological properties, wind speed, air temperature. These factors may affect the implications of the results. For instance, in different seasons, the soil properties, especially soil microorganisms, may largely change and lead to the variation in N2O emission regardless of rainy or fine days. Soil moisture, temperature, and redox data are critical to substantiate the claim that rain-induced anoxia drives denitrification. Their absence weakens causal inferences.**

We have discussed the influence of seasons on the variation of $N_2O$ emission in the manuscript (Lines 293-296 and Lines 299-301). The key soil physical/chemical parameters relevant to determine the pathways of $N_2O$ production in soils, such as bulk density and the concentrations of $NH_4^+$, $NO_3^-$, and $NO_2^-$ in soils, were presented in the manuscript (Lines 105-107) and supplement (Text S1 and Table S1).

In addition, we also have added soil moisture (WFPS) in the manuscript (Section 4.3) and the supplement (Text S1 and Table S1) and included a discussion of redox conditions supporting that rain-induced anoxia drives denitrification (Lines 466-470).

Finally, in response to your request, while the soil microbiological properties and redox data were unavailable in this study, we would like to add the wind speed and air temperature data to Table S1 in the supplement (as follows).

| Soil type | Time | T# | Wind speed | P# | WFPS# | [NH$_4^+$] | [NO$_3^-$] | [NO$_2^-$] | Flux-N$_2$O | δ$^{18}$O (NO$_2^-$) | Δ$^{17}$O (NO$_2^-$) | δ$^{15}$N (N$_2$O) | δ$^{18}$O (N$_2$O) | Δ$^{17}$O (N$_2$O) |
|---|---|---|---|---|---|---|---|---|---|---|---|---|---|---|
| | | °C | m / s | mm | % | | mg N kg$^{-1}$ | | μg N m$^{-2}$ h$^{-1}$ | | | ‰ | | |
| Natural soil | 2022/4/26 | 22.3 | 5.3 | 0 | 71.6 | 11.5 | 1.2 | 0.03 | 3.6 | 12.03 | 0.50 | -27.5 | 26.1 | -0.32 |
| | 2022/6/9 | 25.2 | 4.8 | 0 | 60.5 | 7.6 | 0.9 | 0.01 | 0.6 | 6.72 | 0.04 | - | - | - |
| | 2022/7/11 | 30.5 | 3.7 | 0 | 77.4 | 10.1 | 0.4 | 0.16 | 6.9 | 5.19 | 0.25 | -17.9 | 37.6 | -0.40 |
| | 2022/8/8 | 30.2 | 3.6 | 17.5 | 61.1 | 8.9 | 0.4 | 0.17 | 6.9 | 6.98 | 0.29 | -26.6 | 18.4 | 0.17 |
| | 2022/9/8 | 26.6 | 2.1 | 11.5 | 92.3 | 9.5 | 0.5 | 0.09 | 23.7 | 7.37 | 0.06 | -19.5 | 30.9 | -0.06 |
| | 2022/9/13 | 31.1 | 3.3 | 0 | 69.7 | 12.5 | 1.6 | 0.12 | 6.0 | 2.42 | 0.13 | -21 | 33.2 | -0.28 |
| | 2022/10/13 | 20.3 | 1.5 | 0 | 60.9 | 16.9 | 1.6 | 0.21 | 6.5 | 3.10 | 0.09 | -21.3 | 27.6 | -0.34 |
| | 2022/11/5 | 17.4 | 3.7 | 0 | 59.6 | 0.7 | 5.9 | 0.03 | 1.1 | 4.51 | 0.21 | -21.2 | - | - |
| | 2022/12/14 | 6.4 | 7.0 | 0 | 63.7 | 9.7 | 2.2 | 0.15 | 0.8 | 5.84 | 0.11 | -21.1 | - | -0.31 |
| | 2023/1/29 | 5.3 | 3.0 | 0 | 74.3 | 8.5 | 2.5 | 0.16 | -0.2 | 6.22 | 0.24 | - | - | - |
| | 2023/3/9 | 18.9 | 4.2 | 0 | 68.7 | 8.6 | 6.0 | 0.12 | 2.4 | 5.55 | 0.25 | -22.4 | 26.6 | -0.26 |
| | 2023/3/23 | 18.4 | 3.9 | 16.5 | 91.5 | 13.0 | 3.2 | 0.45 | 67.3 | 5.93 | 0.29 | -25.9 | 22.7 | 0.26 |
| | 2023/4/7 | 16.3 | 5.8 | 32.5 | 113.7 | 11.7 | 1.2 | 0.16 | 77.4 | 6.91 | 0.23 | -18.5 | 28.2 | 0.22 |
| | 2023/4/11 | 19.9 | 5.2 | 0 | 66.2 | 11.6 | 1.1 | 0.23 | 9.8 | 6.85 | 0.20 | -21.7 | 33.4 | -0.11 |
| | 2023/4/15 | 13.7 | 1.9 | 33.5 | 108.4 | 11.7 | 0.9 | 0.19 | 20.0 | 4.24 | 0.25 | -18.0 | 31.1 | 0.18 |
| | 2023/5/17 | 31.2 | 2.9 | 0 | 61.7 | 10.1 | 0.7 | 0.13 | 3.7 | 5.75 | 0.40 | -25.3 | 31.9 | -0.34 |
| | 2023/6/2 | 21.4 | 2.3 | 137 | 106.7 | 6.2 | 0.03 | 0.04 | 37.4 | 5.79 | 0.19 | -13.8 | 36.2 | -0.03 |
| | 2023/7/4 | 31.1 | 4.4 | 0 | 58.7 | 7.6 | 0.1 | 0.15 | 3.8 | 6.25 | 0.40 | -25.8 | - | -0.39 |
| Fertilized soil | 2023/7/18 NF | 34.6 | 4.1 | 0 | 71.9 | 12.4 | 2.0 | 0.20 | 5.2 | 2.69 | 0.42 | -17.1 | 36.1 | -0.37 |
| | 2023/7/22 NF | 30.9 | 4.7 | 0 | 59.4 | 12.0 | 2.6 | 0.26 | 4.2 | 1.33 | 0.35 | -12.2 | 40 | -0.32 |
| | 2023/7/18 U | 34.6 | 4.1 | 0 | 80.3 | 410.2 | 5.4 | 0.10 | 70.6 | 7.64 | 0.31 | -39.3 | 34.4 | -0.14 |
| | 2023/7/22 U | 30.9 | 4.7 | 0 | 62.9 | 435.9 | 20.5 | 0.07 | 56.7 | 5.40 | 0.17 | -33.3 | 25.7 | -0.16 |
| | 2023/7/18 CS | 34.6 | 4.1 | 0 | 47.6 | 12.9 | 247.8 | 0.09 | 112.3 | 28.98 | 8.26 | -19.3 | 54.1 | 8.22 |
| | 2023/7/22 CS | 30.9 | 4.7 | 0 | 37.9 | 18.7 | 309.0 | 0.07 | 39.4 | 45.24 | 12.32 | -11.3 | 58.7 | 7.36 |

T#: Air temperature; P#: Precipitation; WFPS#: Water-filled pore space

**Variability in N₂O fluxes (e.g., ±28.0 µg N m⁻² h⁻¹ on rainy days) warrants discussion (e.g., soil heterogeneity, rain intensity).**

While the present results show no significant relationships between N₂O flux and soil moisture (WFPS), precipitation amount, temperature, and wind speed on rainy days (Figures R1a, R1b, R1c, and R1d), we recognize that further rainy-day monitoring, incorporating assessment of factors such as soil heterogeneity and rain intensity, will be needed in the future to explain the observed variability in N₂O flux on rainy days. We would like to add this information to the revised manuscript.

[Figure]

**Figure R1.** The flux of N₂O on rainy days plotted as a function of the WFPS (a), that of amount of precipitation (b), that of air temperature (c), and that of wind speed (d).

**3. The Δ¹⁷O of N₂O on rainy days (+0.12‰) is lower than that of NO₂⁻ (+0.23‰). The authors should address whether this reflects mixing of oxygen sources (e.g., H₂O, O₂) or kinetic fractionation during denitrification.**

We have quantitatively assessed the effect of kinetic fractionation during denitrification on the Δ¹⁷O of N₂O in the manuscript (Lines 388-398) and concluded that the possible range of variations in the Δ¹⁷O value of N₂O from that of NO₂⁻ to be less than 0.075 ‰. Thus, the lower Δ¹⁷O of N₂O on rainy days (+0.12 ‰) compared to

that of $NO_2^-$ (+0.23 ‰) mainly reflects mixing of oxygen sources derived from $O_2$ (−0.44 ‰).

**On fine days, the $\Delta^{17}O$ of $N_2O$ (−0.30‰) differs from $O_2$ (−0.44‰). Potential contributions from $H_2O$ ($\Delta^{17}O \approx 0$‰) during nitrification should be discussed.**

We have already discussed the potential contributions from $H_2O$ during nitrification (Lines 445-454) and concluded that the contribution of O atoms derived from soil $H_2O$ was minor during the oxidation of $NH_4^+$ to produce $N_2O$.

**4. A more in depth comparison with complementary isotopic compositions (e.g., $\delta^{15}N$, $\delta18O$) would strengthen pathway discrimination.**

We have already compared the $\delta^{15}N$ and $\delta^{18}O$ of $N_2O$ with $\Delta^{17}O$ for the pathway discrimination in the manuscript as follows.

Although the $\delta^{18}O$ values of $N_2O$ emitted from the soil were significantly higher than those of the sources of O atoms in $N_2O$ ($NO_2^-$, $O_2$, and $H_2O$; Figures 4e and 6a) due to the fractionations of oxygen isotopes during the production and/or reduction of $N_2O$, the $\Delta^{17}O$ values of $N_2O$ remained within the range of these sources. This indicates that $\Delta^{17}O$ primarily reflects the pathways of $N_2O$ production, providing information distinct from the $\delta^{18}O$ signature because $\Delta^{17}O$ is stable during the processes of biogeochemical isotope fractionation (Lines 517-523).
Moreover, while $N_2O$ emission from the forested soil did not show significant differences in $\delta^{15}N$ and $\delta^{18}O$ values between fine and rainy days due to the fractionations of nitrogen and oxygen isotopes (Figures 4f and 4h), the significant difference in the $\Delta^{17}O$ values of $N_2O$ between fine and rainy days (Figure 4d) highlights $\Delta^{17}O$ to be a promising natural signature for identifying the pathways of $N_2O$ production in soils (Lines 524-528).

**Specific comments:**
**1. Line 56, 59, spell out "SP" for its first appearance**

Thank you for your comment. We would like to spell out "SP" (site preference) in the revised manuscript

**2. Line 150, which kind of autoanalyzer**

Thank you for your comment. We would like to revise the sentence as follows.
Their concentrations were determined using a high performance microflow analyzer (QuAAtro 39 Autoanalyzer, BLTEC, Osaka, Japan) (Lines 149-151).

**3. Line 205, spell out "VSMOW" for its first appearance**

Thank you for your comment. We would like to spell out "VSMOW" (Vienna Standard Mean Ocean Water) in the revised manuscript

**4. Line 357, Identification of pathways of N2O production in forested soil using Δ17O signature, the subhead can be changed to "Identification of N2O production pathways in forested soil using Δ17O signature.**

Thank you for your comment. We would like to change the subhead to "Identification of $N_2O$ production pathways in forested soil using $\Delta^{17}O$ signature" in the revised manuscript.

**5. Section 4.2, this section only reports experimental results and has no discussion.**

In Section 4.2, in addition to presenting experimental results, we have also included discussion regarding the possible contributions of O atoms to soil $N_2O$ derived from soil $H_2O$ during denitrification and nitrification processes.

**6. The figures appear to be crudely constructed, seemingly pieced together, with text added afterward. Subfigures are misaligned, and axis labels are inconsistently positioned. It is recommended to use professional illustration tools to improve the clarity and precision of the figures.**

Thank you for your suggestion. We would like to carefully revise all figures using professional illustration software to ensure proper alignment of subfigures, consistent positioning of axis labels, and overall improved visual clarity in the manuscript.

We would like to thank you for the helpful comments. We hope that our responses to your comments are satisfactory.

Sincerely,
Weitian Ding
Graduate School of Environmental Studies,
Nagoya University
Furo-cho, Chikusa-ku, Nagoya,
464-8601, JAPAN
E-mail: dwt530754556@gmail.com
Cc: Drs. Urumu Tsunogai and Fumiko Nakagawa

---

## Author Response (AR1)

June 15, 2025

Dr. David McLagan
Editor of Biogeosciences

Title: Triple oxygen isotope evidence for the pathway of nitrous oxide production in a forested soil with increased emission on rainy days
Authors: Weitian Ding et al.
MS No.: egusphere-2025-996

Dear Dr. David McLagan:

Thank you very much for handling our manuscript. We would like to thank the referees as well for the constructive comments on our manuscript.

We have carefully studied the comments and revised the manuscript accordingly. We include below point-by-point responses to the comments, and detailed descriptions of the modifications we made to the manuscript. Besides, we also uploaded the revised manuscript in MS Word, in which all the revisions from BGD version were recorded.

Specifically, we have incorporated clarifications in response to Reviewer 1's comments [1], [3], [6], and [8] into the revised manuscript. Additionally, we have implemented some of Reviewer 2's comments throughout the revised manuscript.

We hope that with these changes you will find our revised manuscript appropriate for publication in your journal.

Sincerely yours,
Weitian Ding
Postdoctoral researcher
Graduate School of Environmental Studies,
Nagoya University
Furo-cho, Chikusa-ku, Nagoya,
464-8601, JAPAN
E-mail: dwt530754556@gmail.com
Cc: Drs. Urumu Tsunogai and Fumiko Nakagawa

**Response to the referee #1:**

**[1] Section 2.3: While extraction with 2 M KCl is standard for measuring extractable soil nitrate and ammonium, significant loss of soil nitrite can occur during the extraction. This issue is well recognized in the soil nitrogen cycling community (e.g., Homyak et al., 2015). Given that nitrite concentration and Δ$^{17}$O measurements are critical to this study, could the authors discuss how potential nitrite loss and associated isotopic fractionation might impact their analysis?**

While the 2M KCl extraction is widely used for soil nitrite ($NO_2^-$) analysis (e.g., Lewicka-Szczebak et al., 2021; Shen et al., 2003), Homyak et al. (2015) raised the concerns that the recovery of soil $NO_2^-$ could be low when using KCl solutions compared to deionized water.

To evaluate this potential issue, we conducted a comparative experiment in April 2022 prior to this study. We collected a soil sample from our study site, which was thoroughly homogenized and divided into two 50 g subsamples. Each subsample was then extracted with either 50 mL of 2M KCl solution or 50 mL MQ water, following the same analytical procedures used in this study.

Our results showed consistent values between the two extraction methods: the KCl-extracted sample yielded a $NO_2^-$ concentration of 0.90 μM with Δ$^{17}$O of 0.55±0.1‰, while the MQ water-extracted sample showed a $NO_2^-$ concentration of 0.98 μM with Δ$^{17}$O of 0.62±0.1‰. Because both the concentration and Δ$^{17}$O value of soil $NO_2^-$ in KCl solution and MQ water showed no significant differences, we concluded that for our soil type and experimental conditions, the use of 2M KCl solution introduced negligible bias in terms of $NO_2^-$ recovery or Δ$^{17}$O measurements compared to MQ water extraction.

We emphasized this in the revised manuscript (P13/L276-283) and supplement (P2/L36-52).

**[2] Section 2.4: The manuscript uses a β (triple oxygen proportionality factor) range of 0.525 to 0.5305 to quantify the potential impact on Δ$^{17}$O. Although several references are cited, it is unclear why this specific range was chosen. Please clarify. In particular, earlier studies (e.g., Matsuhisa et al., 1978; summarized by Miller, 2002) reported lower β values (e.g., ~0.5164). How would using lower β values affect the results?**

Our selection of this range was based on evidence from recent experimental and theoretical studies examining oxygen isotope fractionation across various compounds (CO, $O_2$, NO, $CO_2$, $NO_2$, $H_2O$, $SO_2$, $SO_3$, $CO_3^{2-}$, and $SiO_2$), as documented by (Cao and Liu, 2011; Pack and Herwartz, 2014; Sharp and Wostbrock, 2021). These studies collectively demonstrate that this range encompasses most equilibrium and kinetic fractionation processes. Thus, for the calculation of Δ$^{17}$O value of $N_2O$, we adopted the midpoint value (β = 0.528) of this range.

Regarding your concern about lower β values reported in earlier work (Matsuhisa et al., 1978), we note that while Matsuhisa et al. (1978) did observe β values as low as 0.5164 in some terrestrial rock and water samples, they ultimately choose 0.52 for the quartz-water system as the most representative value. To address how such lower β values might affect our results, we quantified the possible variations in the Δ$^{17}$O values of $N_2O$ during each reaction using β = 0.52. Our calculations

(following the methodology in Section 4.1 and Figure 7) show that this would introduce variations in $\Delta^{17}O$ values of $N_2O$ (derived from soil $NO_2^-$ and $O_2$) of less than 0.2 ‰ (Figure R1). This potential variation is significantly smaller than the observed $\Delta^{17}O$ difference between $O_2$ and $NO_2^-$ in our forested soil samples (average 0.7‰; Figure 4c). We concluded that even using such low β value (β = 0.52), our key findings or interpretations can't be affected.

[Figure]

**Figure R1.** Schematic showing the possible variations in the $\Delta^{17}O$ value of $N_2O$ from that of the source of O atoms ($O_2$ and $NO_2^-$) during transformations, including nitrification (orange circles), denitrification (green circles), and reduction (yellow circles), due to variations in isotope fractionation and β from 0.520 to 0.5305

**[3] Lines 362–363 and throughout the analysis: A constant $\Delta^{17}O$ value was assumed for $O_2$ (-0.44‰) and soil $H_2O$ (0.03‰). Please clarify whether these values could vary due to hydrological and biogeochemical cycling. For instance, could $O_2$ diffusion and heterotrophic consumption affect $O_2$ $\Delta^{17}O$, or could evaporation significantly alter soil $H_2O$ $\Delta^{17}O$?**

While the $\Delta^{17}O$ values of soil $O_2$ and $H_2O$ used in this study were referred from atmospheric $O_2$ and rainwater, respectively, the processes in soil, including diffusion and respiration of $O_2$ and evaporation and infiltration of rainwater, may cause significant isotopic fractionations of $\delta^{18}O$, which could consequently alter the $\Delta^{17}O$ values of atmospheric $O_2$ and rainwater. Thus, we evaluated the possible variations in the $\Delta^{17}O$ values of $O_2$ and $H_2O$ in soil compared to those of atmospheric $O_2$ and rainwater. The details are presented below.

For soil $O_2$, Aggarwal and Dillon (1998) measured $\delta^{18}O$ values in soil gas at a depth of 3-4 m at a site near Lincoln, Nebraska, USA ranged from +23.3 ‰ to +27.2 ‰ (Table R1), showing the values were comparable with that of atmospheric $O_2$ (+23.5 ‰ after adjustment in Aggarwal and Dillon. 1998). This confirms that the isotopic fractionation of soil $O_2$ induced from soil respiration and diffusion processes wasn't significant. Because the maximum variation in $\delta^{18}O$ from atmospheric $O_2$ to soil $O_2$ was less than 3.7 ‰ (27.2 ‰ − 23.5 ‰), using the method presented in

Section 4.1 and Figure 7, we quantified the possible variations in the $\Delta^{17}O$ value of soil $O_2$ from that of atmospheric $O_2$ to be less than 0.01 ‰. Thus, we ignored the negligible variations in the manuscript.

Similarly, for soil $H_2O$, Lyu (2021) observed that $\delta^{18}O$ values in soil $H_2O$ at the depths of 0-5 cm, 15-20 cm, and 40-45 cm in a subtropical forest plantation ranged from −4 ‰ to −10 ‰ (Figure R2), which fully overlapped with local rainwater (−1 ‰ to −16 ‰), indicating insignificant isotopic fractionations of soil $H_2O$ during hydrological process such as infiltration and evaporation compared to rainwater. Besides, Aron et al. (2021) compiled $\Delta^{17}O$ values of terrestrial $H_2O$ including rainwater, surface and subsurface water in earth, ranged from +0.06 to −0.06 ‰ and didn't show significant difference with each other, which also indicating that the possible variations of $\Delta^{17}O$ values of soil $H_2O$ compared to that of rainwater should be negligible. Finally, we added the variations of $\Delta^{17}O$ values (+0.06 / −0.06 ‰) of terrestrial $H_2O$ reported in Aron et al. (2021) to Figures 4 and 6 as the uncertainties of $\Delta^{17}O$ values of soil $H_2O$ in the revised manuscript.

We added this information into the revised manuscript (P20-21/L424-451).

**Table R1.** Concentration and isotopic compositions of soil gas oxygen and carbon dioxide from the midwestern USA site (Aggarwal and Dillon. 1998).

| Location | Depth (m) | Date | $CO_2$ (%) | $O_2$ (%) | $\delta^{18}O\text{-}O_2$[a] ‰, SMOW | $\delta^{13}C\text{-}CO_2$[a] ‰, PDB |
|---|---|---|---|---|---|---|
| 1b | 2.9 | June 94 | 1.8 | 13.8 | 27.2 | −21.8 |
| | 2.9 | Dec. 94 | 2.3 | 15.1 | 24.2 | −21.7 |
| 2b | 3.4 | June 94 | 0.7 | 16.3 | 25.1 | −22.4 |
| 2b | 3.4 | Dec. 94 | 0.9 | 15.2 | 23.3 | −22.8 |
| 3b | 4.1 | June 94 | 0.7 | 15.7 | 25.3 | −22.1 |
| 3b | 4.1 | Dec. 94 | 1.4 | 15.2 | 25.1 | −24.0 |
| 5b | 3.2 | June 94 | 0.8 | 17.6 | 24.7 | −21.7 |
| 5b | 3.2 | Dec. 94 | 1.0 | 16.3 | 24.7 | −21.0 |
| 4b | 3.2 | June 94 | 2.3 | 17.1 | 24.5 | −20.0 |
| 4b | 3.2 | Dec. 94 | 3.0 | 16.0 | 24.0 | −19.9 |

[a] Isotopic values reported are averages of duplicate analyses with a standard deviation of 0.3‰ for $\delta^{18}O$ and 0.2‰ for $\delta^{13}C$. The oxygen isotope ratios have been adjusted to an atmospheric oxygen $\delta^{18}O$ of 23.5 ‰.

[Figure]

**Figure R2.** Temporal variations of the amount of precipitation, $\delta^{18}O$ in precipitation, and weighted average $\delta^{18}O$ in soil water source during winter, spring, summer, and autumn, at 0–5 (0–5 cm), 15–

20 (15–20 cm), and 40–45 (40–45 cm) cm depths (Lyu. 2021).

**[4] Lines 378–397: The characterization of δ¹⁸O offsets between O₂ and N₂O, and between NO₂⁻ and N₂O, does not necessarily represent true isotope effects between N₂O and its oxygen precursors because field-measured N₂O is a mixture of multiple sources. For example, in Fig. 6a, the actual δ¹⁸O difference between NO₂⁻ and N₂O may be larger than calculated if O₂-derived N₂O has δ¹⁸O values similar to that of O₂. Similarly, the O₂-N₂O difference may be smaller than estimated. This mixing effect could confound the use of δ¹⁸O differences to estimate Δ¹⁷O variations and warrants further clarification.**

You mentioned that the true field-measured N₂O is a mixture of multiple sources is correct. However, the mixture ratios of N₂O produced through nitrification and denitrification were unknown. Thus, in our theoretical calculations for the possible variations in the $\Delta^{17}O$ values of N₂O in Section 4.2, we separated the nitrification and denitrification to discuss the possible variations.

After we supposed that if all O atoms in N₂O were derived from O₂, the average in $\delta^{18}O$ from O₂ to N₂O due to nitrification ($\Delta\delta^{18}O$ (N₂O−O₂)) was estimated to be 9 ‰ on average. Similarly, after we supposed that if all O atoms in N₂O were derived from NO₂⁻, the average variation in $\delta^{18}O$ from NO₂⁻ to N₂O due to fractionation ($\Delta\delta^{18}O$(N₂O−NO₂⁻)) was estimated to be 25 ‰ on average.

**[5] Fig. 7 and related discussion: I commend the authors for conducting a sensitivity analysis to assess how much Δ¹⁷O variation may stem from biogeochemical processes versus purely geochemical processes (i.e., β variability). However, applying the β range to the net δ¹⁸O difference between N₂O and oxygen sources treats the N₂O-producing processes as a single step. In reality, processes like nitrite reduction involve multiple sub-steps (e.g., NO₂⁻ to NO, NO to N₂O, isotope exchange with H₂O), each potentially associated with different β values. This could lead to larger Δ¹⁷O variations than those estimated from a single-step approach. This limitation should be discussed.**

Thank you for your comment. Because the β values for processes of NO₂⁻ to NO and NO to N₂O should be included in the range of 0.525 to 0.5305 (Cao and Liu, 2011; Matsuhisa et al., 1978; Pack and Herwartz, 2014; Sharp and Wostbrock, 2021), the processes of NO₂⁻ to NO and NO to N₂O were merged into the process of denitrification for the theoretical calculations for the possible variations in the $\Delta^{17}O$ values of N₂O. As a result, the calculated possible variations in $\Delta^{17}O$ during denitrification (less than 0.075 ‰) incorporated the β variability across these sub-steps.

Besides, because the estimated $\Delta^{17}O$ values of soil NO₂⁻ included the effect of oxygen isotope exchange between soil NO₂⁻ and H₂O, the oxygen isotope exchange between soil NO₂⁻ and H₂O can't affect the $\Delta^{17}O$ values of N₂O.

Additionally, because we concluded the contributions of O atoms derived from soil H₂O were minor during the reduction of NO₂⁻ and oxidation of NH₄⁺ to produce N₂O, the discussion for possible variations in $\Delta^{17}O$ values of N₂O due to the process of the contribution of O atoms derived from soil H₂O was ignored.

**[6] Lines 434–438: It is unclear how the 24% contribution of soil H₂O was derived.**

This calculation was based on isotopic mass balance. In the plot fertilized with CS, the average $\Delta^{17}O$ value of $N_2O$ emitted from the soil 2 and 6 days after the fertilization was +7.79 ‰. The $\Delta^{17}O$ value of the possible source of O atoms in $N_2O$ was +10.30 ‰ for soil $NO_2^-$, +0.03 ‰ for soil $H_2O$, and −0.44 ‰ for $O_2$, respectively. If all the O atoms with low $\Delta^{17}O$ values in $N_2O$ were derived from soil $H_2O$ (+0.03 ‰) in the CS plot, the contribution of O atoms derived from soil $H_2O$ was calculated to be 24 % ((10.30 ‰ – 7.79 ‰) / (10.30 ‰ – 0.03 ‰)). If the $O_2$ also contributed to the $N_2O$ production in the CS plot, the contribution of O atoms derived from soil $H_2O$ should be further reduced. As a result, we determined that the maximum possible contribution of O atoms derived from soil $H_2O$ during the reduction of $NO_2^-$ to $N_2O$ was 24 %.

We clarified that in the revised manuscript (P22/L476-482).

**Additionally, Fig. 6b shows that the $\Delta^{17}O$ of $N_2O$ in the CS plot was significantly lower than that of $NO_2^-$ six days after tracer addition. This suggests that soil $H_2O$ may have played a significant role during nitrite reduction to $N_2O$.**

Compared to the value observed 2 days after fertilization, the $\Delta^{17}O$ value of $N_2O$ emitted from the soil in the CS plot 6 days after fertilization became lower than that of soil $NO_2^-$ (Figure 6b), implying that (1) the soil $H_2O$ have played a significant role 6 days after fertilization as suggested, or (2) the relative contribution of nitrification to $N_2O$ production increased 6 days after fertilization. Because the main pathway to produce $N_2O$ was nitrification in the NF plot (no fertilizer addition) (Figure 6b), the diminishing fertilization effect over time resulted in reduced $N_2O$ production through denitrification was responsible for the relative contribution of nitrification to $N_2O$ production increased 6 days after fertilization. The significant decrease in $N_2O$ flux from 112.3 to 39.4 μg N m$^{-2}$ h$^{-1}$ between 2 and 6 days after fertilization further confirm the diminishing fertilization effect over time.

Importantly, similar to 2 days after fertilization, the $\Delta^{17}O$ value of $N_2O$ emitted from the soil in the CS plot 6 days after fertilization (+7.36 ‰) remained closer to that of soil $NO_2^-$ (+12.32 ‰) than that of atmospheric $O_2$ (−0.44 ‰) and $H_2O$ (+0.03 ‰), consistent with our conclusion that the denitrification became the main pathway of $N_2O$ production in the CS plot.

**[7] Lines 444–449 and Fig. 6b: Apparent differences in $\Delta^{17}O$ between soil $H_2O$ and $N_2O$ cannot be used to conclusively rule out $H_2O$ contributions during $N_2O$ production. In the NF and U plots, the $\Delta^{17}O$ of soil $H_2O$ lies between that of $NO_2^-$ and $N_2O$, and both soil $H_2O$ and $NO_2^-$ have higher $\Delta^{17}O$ than $O_2$. Could significant $H_2O$ exchange during $N_2O$ production explain these observations, leading to a mixed $\Delta^{17}O$ signal from both $H_2O$- and $O_2$-derived $N_2O$?**

In NF plot, the average $\Delta^{17}O$ value of $N_2O$ (−0.35 ‰) measured 2 and 6 days after fertilization was close to that of $O_2$ (−0.44 ‰) compared to that of soil $H_2O$ (+0.03 ‰) and soil $NO_2^-$ (+0.38 ‰) (Figure 6b), implying that the O atoms in $N_2O$ mainly derived from $O_2$ rather than soil $H_2O$. Thus, the $H_2O$ contribution during $N_2O$ production can't be significant in this case.

In U plot, while the significant $H_2O$ contribution during $N_2O$ production could explain the $\Delta^{17}O$ value of $N_2O$ becoming higher than that in NF plot after fertilization, the observed increases in the emission flux of $N_2O$ from the soil in NF plot (from 4.7 to 63.7 μg N m$^{-2}$ h$^{-1}$; Table S1 in supplement) can't be explained by the significant $H_2O$ contribution during $N_2O$ production. Thus, we maintain our conclusion that the increase in $N_2O$ production through $NO_2^-$ reduction was responsible for the $\Delta^{17}O$ values of $N_2O$ produced in the U plot in response to fertilization of urea/$NH_4^+$.

**[8] Section 4.5: Early in the manuscript, the authors argue that bulk isotopic and SP-based techniques for $N_2O$ source apportionment are limited due to isotopic fractionations during cycling (lines 56-61), whereas $\Delta^{17}O$ measurements may be more robust. After presenting the results, I would encourage the authors to revisit this point with more specificity. Given potential complications such as $H_2O$ exchange and multiple contributing sources ($H_2O$, $O_2$, $NO_2^-$), can $\Delta^{17}O$ measurements realistically achieve quantitative source apportionment? If so, what would the total uncertainty be, considering analytical precision, β variability, and uncertainties from the Keeling approach? Under what conditions would $\Delta^{17}O$ approaches be preferable to conventional methods, and when might they be less effective?**

We added the uncertainty information for using $\Delta^{17}O$ as a natural signature for identifying $N_2O$ production pathways including $H_2O$ contributions and β variability in Section 4.5 in the revised manuscript (P26-27/L575-579).

**Reference**

Aggarwal, P. K. and Dillon, M. A.: Stable Isotope Composition of Molecular Oxygen in Soil Gas and Groundwater: A Potentially Robust Tracer for Diffusion and Oxygen Consumption Processes, Geochimica et Cosmochimica Acta, 62, 577–584, https://doi.org/10.1016/S0016-7037(97)00377-3, 1998.

Aron, P. G., Levin, N. E., Beverly, E. J., Huth, T. E., Passey, B. H., Pelletier, E. M., Poulsen, C. J., Winkelstern, I. Z., and Yarian, D. A.: Triple oxygen isotopes in the water cycle, Chemical Geology, 565, 120026, https://doi.org/10.1016/j.chemgeo.2020.120026, 2021.

Cao, X. and Liu, Y.: Equilibrium mass-dependent fractionation relationships for triple oxygen isotopes, Geochimica et Cosmochimica Acta, 75, 7435–7445, https://doi.org/10.1016/j.gca.2011.09.048, 2011.

Homyak, P. M., Vasquez, K. T., Sickman, J. O., Parker, D. R., and Schimel, J. P.: Improving Nitrite Analysis in Soils: Drawbacks of the Conventional 2 M KCl Extraction, Soil Science Society of America Journal, 79, 1237–1242, https://doi.org/10.2136/sssaj2015.02.0061n, 2015.

Lewicka-Szczebak, D., Jansen-Willems, A., Müller, C., Dyckmans, J., and Well, R.: Nitrite isotope characteristics and associated soil N transformations, Sci Rep, 11, 5008, https://doi.org/10.1038/s41598-021-83786-w, 2021.

Lyu, S.: Variability of $\delta^2H$ and $\delta^{18}O$ in Soil Water and Its Linkage to Precipitation in an East Asian Monsoon Subtropical Forest Plantation, Water, 13, 2930, https://doi.org/10.3390/w13202930, 2021.

Matsuhisa, Y., Goldsmith, J. R., and Clayton, R. N.: Mechanisms of hydrothermal crystallization of quartz at 250°C and 15 kbar, Geochimica et Cosmochimica Acta, 42, 173–182,

https://doi.org/10.1016/0016-7037(78)90130-8, 1978.

Pack, A. and Herwartz, D.: The triple oxygen isotope composition of the Earth mantle and understanding $\Delta^{17}O$ variations in terrestrial rocks and minerals, Earth and Planetary Science Letters, 390, 138–145, https://doi.org/10.1016/j.epsl.2014.01.017, 2014.

Sharp, Z. D. and Wostbrock, J. A. G.: Standardization for the Triple Oxygen Isotope System: Waters, Silicates, Carbonates, Air, and Sulfates, Reviews in Mineralogy and Geochemistry, 86, 179–196, https://doi.org/10.2138/rmg.2021.86.05, 2021.

Sharp, Z. D., Gibbons, J. A., Maltsev, O., Atudorei, V., Pack, A., Sengupta, S., Shock, E. L., and Knauth, L. P.: A calibration of the triple oxygen isotope fractionation in the $SiO_2$–$H_2O$ system and applications to natural samples, Geochimica et Cosmochimica Acta, 186, 105–119, https://doi.org/10.1016/j.gca.2016.04.047, 2016.

Shen, Q. R., Ran, W., and Cao, Z. H.: Mechanisms of nitrite accumulation occurring in soil nitrification, Chemosphere, 50, 747–753, https://doi.org/10.1016/S0045-6535(02)00215-1, 2003.

Uechi, Y. and Uemura, R.: Dominant influence of the humidity in the moisture source region on the $^{17}O$-excess in precipitation on a subtropical island, Earth and Planetary Science Letters, 513, 20–28, https://doi.org/10.1016/j.epsl.2019.02.012, 2019.

**Response to the referee #2:**

**1. The study site is very local. The study site's soil type, vegetation, and climate should be contextualized relative to other ecosystems to assess broader applicability.**

In this study, we focused on monitoring the temporal variations of $\Delta^{17}O$ of soil $N_2O$ to evaluate whether the $\Delta^{17}O$ of $N_2O$ can be a signature for identifying the main pathway of $N_2O$ production.

While the current work emphasizes temporal variations of $\Delta^{17}O$ of soil $N_2O$ at a forested soil, we acknowledge the importance of contextualizing these findings across diverse ecosystems. In futural studies, we plan to investigate the spatial variations of $\Delta^{17}O$ of soil $N_2O$ to access the broader applicability and the connections with the variations of soil type, vegetation, and climate.

**In addition, the sample numbers of N2O gas samples are very limited in this study. For instance, only five data and 18 data are illustrated in Figure 3 and 4.**

In Figure 3, the 5 data points represent an example of changes in the concentration and isotopic compositions ($\delta^{15}N$, $\delta^{18}O$, and $\Delta^{17}O$) of $N_2O$ in gas samples during the observation on September 8, 2022. To maintain conciseness, we presented only a subset of data (5 data) in Figure 3 in the main text, while the complete dataset (89 data points) is provided in the supplement (Figure S5).

In Figure 4, 18 data points were estimated from the complete dataset (89 data) using the Keeling plot approach. Importantly, these 18 data points fully support our key findings: (1) $N_2O$ emitted from the soil exhibited significantly higher $\Delta^{17}O$ values on rainy days (+0.12±0.13 ‰) than on fine days (−0.30±0.09 ‰) and (2) the emission flux of $N_2O$ was significantly higher on rainy days (38.8±28.0 μg N $m^{-2}h^{-1}$) than on fine days (3.8±3.1 μg N $m^{-2}h^{-1}$).

**The confidence degree of the line fitting and the representative of the results should be further clarified.**

We added the confidence degree of the line fitting in Figure 3 in the revised manuscript as follows. We have emphasized that Figure 3 represent an example of changes in the concentration and isotopic compositions of $N_2O$ in gas samples during the observation on September 8, 2022 (P40/L939-943).

[Figure]

**The novelty of the findings and this study should be further highlighted by comparing with prior soil $\Delta^{17}O$ studies.**

We highlighted the novelty of this study by comparing it with the prior soil $\Delta^{17}O$ study in the revised manuscript (P5/L86-96) as follows.

Previous studies have identified the elevated $\Delta^{17}O$ values in atmospheric $N_2O$ ($\Delta^{17}O \approx +0.9$ ‰), observed in both stratospheric and tropospheric air (Cliff et al., 1999; Kaiser et al., 2003; Thiemens and Trogler, 1991). Komatsu et al. (2008) subsequently conducted the first $\Delta^{17}O$ measurements of $N_2O$ emitted from a soil to assess whether soil $N_2O$ could be the source of elevated $\Delta^{17}O$ values of atmospheric $N_2O$. However, the temporal variations of the $\Delta^{17}O$ values for $N_2O$ emitted from soil remain unknown. Besides, whether $\Delta^{17}O$ values of $N_2O$ can be used to identify the pathways of $N_2O$ production in soils has not been discussed. Additionally, the advantages of $\Delta^{17}O$ signature, relative to other natural stable isotopes, for identifying the pathways of $N_2O$ production remain unclear. To address these, in this study, we measured precise $\Delta^{17}O$ values for $N_2O$ emitted from forested soil and those for $NO_2^-$ in the soil.

**2. The sample information is missing. The definition of "fine days" and "rainy days" (e.g., precipitation threshold, duration) must be clarified to ensure reproducibility.**

In the original manuscript, we have already defined the fine days and rainy days as follows. A fine day is defined as a day without precipitation for 48 hours prior to the end of each sampling. The total precipitation within 12 h at the end of each sampling of the rainy days exceeded 12 mm.

**In addition to weather conditions (fine or rainy), the other influencing factors are not**

**considered and discussed, for instance, soil physical, chemical and microbiological properties, wind speed, air temperature. These factors may affect the implications of the results. For instance, in different seasons, the soil properties, especially soil microorganisms, may largely change and lead to the variation in N2O emission regardless of rainy or fine days. Soil moisture, temperature, and redox data are critical to substantiate the claim that rain-induced anoxia drives denitrification. Their absence weakens causal inferences.**

We have discussed the influence of seasons on the variation of $N_2O$ emission in the original manuscript (Lines 293-296 and Lines 299-301). The key soil physical/chemical parameters relevant to determine the pathways of $N_2O$ production in soils, such as bulk density and the concentrations of $NH_4^+$, $NO_3^-$, and $NO_2^-$ in soils, were presented in the original manuscript (Lines 105-107) and supplement (Text S1 and Table S1).

In addition, we also have added soil moisture (WFPS) in the manuscript (Section 4.3) and the supplement (Text S1 and Table S1) and included a discussion of redox conditions supporting that rain-induced anoxia drives denitrification.

Finally, in response to your request, while the soil microbiological properties and redox data were unavailable in this study, we added the wind speed and air temperature data to Table S1 in the revised supplement (as follows).

| Soil type | Time | T[#] | Wind speed | P[#] | WFPS[#] | [$NH_4^+$] | [$NO_3^-$] | [$NO_2^-$] | Flux-$N_2O$ | $\delta^{18}O$ ($NO_2^-$) | $\Delta^{17}O$ ($NO_2^-$) | $\delta^{15}N$ ($N_2O$) | $\delta^{18}O$ ($N_2O$) | $\Delta^{17}O$ ($N_2O$) |
|---|---|---|---|---|---|---|---|---|---|---|---|---|---|---|
| | | °C | m / s | mm | % | | mg N kg$^{-1}$ | | µg N m$^{-2}$ h$^{-1}$ | | | ‰ | | |
| Natural soil | 2022/4/26 | 22.3 | 5.3 | 0 | 71.6 | 11.5 | 1.2 | 0.03 | 3.6 | 12.03 | 0.50 | -27.5 | 26.1 | -0.32 |
| | 2022/6/9 | 25.2 | 4.8 | 0 | 60.5 | 7.6 | 0.9 | 0.01 | 0.6 | 6.72 | 0.04 | - | - | - |
| | 2022/7/11 | 30.5 | 3.7 | 0 | 77.4 | 10.1 | 0.4 | 0.16 | 6.9 | 5.19 | 0.25 | -17.9 | 37.6 | -0.40 |
| | 2022/8/8 | 30.2 | 3.6 | 17.5 | 61.1 | 8.9 | 0.4 | 0.17 | 6.9 | 6.98 | 0.29 | -26.6 | 18.4 | 0.17 |
| | 2022/9/8 | 26.6 | 2.1 | 11.5 | 92.3 | 9.5 | 0.5 | 0.09 | 23.7 | 7.37 | 0.06 | -19.5 | 30.9 | -0.06 |
| | 2022/9/13 | 31.1 | 3.3 | 0 | 69.7 | 12.5 | 1.6 | 0.12 | 6.0 | 2.42 | 0.13 | -21 | 33.2 | -0.28 |
| | 2022/10/13 | 20.3 | 1.5 | 0 | 60.9 | 16.9 | 1.6 | 0.21 | 6.5 | 3.10 | 0.09 | -21.3 | 27.6 | -0.34 |
| | 2022/11/5 | 17.4 | 3.7 | 0 | 59.6 | 0.7 | 5.9 | 0.03 | 1.1 | 4.51 | 0.21 | -21.2 | - | - |
| | 2022/12/14 | 6.4 | 7.0 | 0 | 63.7 | 9.7 | 2.2 | 0.15 | 0.8 | 5.84 | 0.11 | -21.1 | - | -0.31 |
| | 2023/1/29 | 5.3 | 3.0 | 0 | 74.3 | 8.5 | 2.5 | 0.16 | -0.2 | 6.22 | 0.24 | - | - | - |
| | 2023/3/9 | 18.9 | 4.2 | 0 | 68.7 | 8.6 | 6.0 | 0.12 | 2.4 | 5.55 | 0.25 | -22.4 | 26.6 | -0.26 |
| | 2023/3/23 | 18.4 | 3.9 | 16.5 | 91.5 | 13.0 | 3.2 | 0.45 | 67.3 | 5.93 | 0.29 | -25.9 | 22.7 | 0.26 |
| | 2023/4/7 | 16.3 | 5.8 | 32.5 | 113.7 | 11.7 | 1.2 | 0.16 | 77.4 | 6.91 | 0.23 | -18.5 | 28.2 | 0.22 |
| | 2023/4/11 | 19.9 | 5.2 | 0 | 66.2 | 11.6 | 1.1 | 0.23 | 9.8 | 6.85 | 0.20 | -21.7 | 33.4 | -0.11 |
| | 2023/4/15 | 13.7 | 1.9 | 33.5 | 108.4 | 11.7 | 0.9 | 0.19 | 20.0 | 4.24 | 0.25 | -18.0 | 31.1 | 0.18 |
| | 2023/5/17 | 31.2 | 2.9 | 0 | 61.7 | 10.1 | 0.7 | 0.13 | 3.7 | 5.75 | 0.40 | -25.3 | 31.9 | -0.34 |
| | 2023/6/2 | 21.4 | 2.3 | 137 | 106.7 | 6.2 | 0.03 | 0.04 | 37.4 | 5.79 | 0.19 | -13.8 | 36.2 | -0.03 |
| | 2023/7/4 | 31.1 | 4.4 | 0 | 58.7 | 7.6 | 0.1 | 0.15 | 3.8 | 6.25 | 0.40 | -25.8 | - | -0.39 |
| Fertilized soil | 2023/7/18 NF | 34.6 | 4.1 | 0 | 71.9 | 12.4 | 2.0 | 0.20 | 5.2 | 2.69 | 0.42 | -17.1 | 36.1 | -0.37 |
| | 2023/7/22 NF | 30.9 | 4.7 | 0 | 59.4 | 12.0 | 2.6 | 0.26 | 4.2 | 1.33 | 0.35 | -12.2 | 40 | -0.32 |
| | 2023/7/18 U | 34.6 | 4.1 | 0 | 80.3 | 410.2 | 5.4 | 0.10 | 70.6 | 7.64 | 0.31 | -39.3 | 34.4 | -0.14 |
| | 2023/7/22 U | 30.9 | 4.7 | 0 | 62.9 | 435.9 | 20.5 | 0.07 | 56.7 | 5.40 | 0.17 | -33.3 | 25.7 | -0.16 |
| | 2023/7/18 CS | 34.6 | 4.1 | 0 | 47.6 | 12.9 | 247.8 | 0.09 | 112.3 | 28.98 | 8.26 | -19.3 | 54.1 | 8.22 |
| | 2023/7/22 CS | 30.9 | 4.7 | 0 | 37.9 | 18.7 | 309.0 | 0.07 | 39.4 | 45.24 | 12.32 | -11.3 | 58.7 | 7.36 |

T[#]: Air temperature; P[#]: Precipitation; WFPS[#]: Water-filled pore space

**Variability in N₂O fluxes (e.g., ±28.0 µg N m⁻² h⁻¹ on rainy days) warrants discussion (e.g., soil heterogeneity, rain intensity).**

While the present results show no significant relationships between N₂O flux and soil moisture (WFPS), precipitation amount, temperature, and wind speed on rainy days (Figures R1a, R1b, R1c, and R1d), we recognize that further rainy-day monitoring, incorporating assessment of factors such as soil heterogeneity and rain intensity, will be needed in the future to explain the observed variability in N₂O flux on rainy days.

[Figure]

**Figure R1.** The flux of N₂O on rainy days plotted as a function of the WFPS (a), that of amount of precipitation (b), that of air temperature (c), and that of wind speed (d).

**3. The Δ¹⁷O of N₂O on rainy days (+0.12‰) is lower than that of NO₂⁻ (+0.23‰). The authors should address whether this reflects mixing of oxygen sources (e.g., H₂O, O₂) or kinetic fractionation during denitrification.**

We have quantitatively assessed the effect of kinetic fractionation during denitrification on the $\Delta^{17}O$ of N₂O in the original manuscript (Lines 388-398) and concluded that the possible range of variations in the $\Delta^{17}O$ value of N₂O from that of NO₂⁻ to be less than 0.075 ‰. Thus, the lower $\Delta^{17}O$ of N₂O on rainy days (+0.12 ‰) compared to that of NO₂⁻ (+0.23 ‰) mainly reflects mixing of oxygen sources derived from O₂ (−0.44 ‰).

**On fine days, the $\Delta^{17}O$ of $N_2O$ (−0.30‰) differs from $O_2$ (−0.44‰). Potential contributions from $H_2O$ ($\Delta^{17}O \approx 0$‰) during nitrification should be discussed.**

We have already discussed the potential contributions from $H_2O$ during nitrification (Lines 445-454 in original manuscript) and concluded that the contribution of O atoms derived from soil $H_2O$ was minor during the oxidation of $NH_4^+$ to produce $N_2O$.

**4. A more in depth comparison with complementary isotopic compositions (e.g., $\delta^{15}N$, $\delta 18O$) would strengthen pathway discrimination.**

We have already compared the $\delta^{15}N$ and $\delta^{18}O$ of $N_2O$ with $\Delta^{17}O$ for the pathway discrimination in original manuscript as follows.

Although the $\delta^{18}O$ values of $N_2O$ emitted from the soil were significantly higher than those of the sources of O atoms in $N_2O$ ($NO_2^-$, $O_2$, and $H_2O$; Figures 4e and 6a) due to the fractionations of oxygen isotopes during the production and/or reduction of $N_2O$, the $\Delta^{17}O$ values of $N_2O$ remained within the range of these sources. This indicates that $\Delta^{17}O$ primarily reflects the pathways of $N_2O$ production, providing information distinct from the $\delta^{18}O$ signature because $\Delta^{17}O$ is stable during the processes of biogeochemical isotope fractionation (Lines 517-523). Moreover, while $N_2O$ emission from the forested soil did not show significant differences in $\delta^{15}N$ and $\delta^{18}O$ values between fine and rainy days due to the fractionations of nitrogen and oxygen isotopes (Figures 4f and 4h), the significant difference in the $\Delta^{17}O$ values of $N_2O$ between fine and rainy days (Figure 4d) highlights $\Delta^{17}O$ to be a promising natural signature for identifying the pathways of $N_2O$ production in soils (Lines 524-528).

**Specific comments:**
1. **Line 56, 59, spell out "SP" for its first appearance**

We spelled out "SP" ($^{15}N$ site preference) in the revised manuscript (P3/L56).

2. **Line 150, which kind of autoanalyzer**

We revised the sentence in the revised manuscript (P8/L153).

3. **Line 205, spell out "VSMOW" for its first appearance**

We spelled out "VSMOW" (Vienna Standard Mean Ocean Water) in the revised manuscript (P9/L184).

4. **Line 357, Identification of pathways of N2O production in forested soil using Δ17O signature, the subhead can be changed to "Identification of N2O production pathways in forested soil using Δ17O signature.**

We changed the subhead to "Identification of $N_2O$ production pathways in forested soil using $\Delta^{17}O$ signature" in the revised manuscript (P17/L369).

**5. Section 4.2, this section only reports experimental results and has no discussion.**

In Section 4.2, in addition to presenting experimental results, we have also included discussion regarding the possible contributions of O atoms to soil $N_2O$ derived from soil $H_2O$ during denitrification and nitrification processes.

**6. The figures appear to be crudely constructed, seemingly pieced together, with text added afterward. Subfigures are misaligned, and axis labels are inconsistently positioned. It is recommended to use professional illustration tools to improve the clarity and precision of the figures.**

We checked the figures including the proper alignment of subfigures, consistent positioning of axis labels, and visual clarity in the revised manuscript.